# Mechanism-centric regulatory network identifies NME2 and MYC programs as markers of Enzalutamide resistance in CRPC

Sukanya Panja[1,8], Mihai Ioan Truica [2,8], Christina Y. Yu [1], Vamshi Saggurthi [1], Michael W. Craige [1], Katie Whitehead[1], Mayra V. Tuiche [1,3], Aymen Al-Saadi[4], Riddhi Vyas[1], Shridar Ganesan[5], Suril Gohel[1], Frederick Coffman[1], James S. Parrott [1], Songhua Quan[2], Shantenu Jha[4], Isaac Kim[5,6], Edward Schaeffer [2], Vishal Kothari [2,9] ✉, Sarki A. Abdulkadir [2,7,9] ✉ & Antonina Mitrofanova [1,5,9] ✉

Heterogeneous response to Enzalutamide, a second-generation androgen receptor signaling inhibitor, is a central problem in castration-resistant prostate cancer (CRPC) management. Genome-wide systems investigation of mechanisms that govern Enzalutamide resistance promise to elucidate markers of heterogeneous treatment response and salvage therapies for CRPC patients. Focusing on the de novo role of MYC as a marker of Enzalutamide resistance, here we reconstruct a CRPC-specific mechanism-centric regulatory network, connecting molecular pathways with their upstream transcriptional regulatory programs. Mining this network with signatures of Enzalutamide response identifies NME2 as an upstream regulatory partner of MYC in CRPC and demonstrates that NME2-MYC increased activities can predict patients at risk of resistance to Enzalutamide, independent of co-variates. Furthermore, our experimental investigations demonstrate that targeting MYC and its partner NME2 is beneficial in Enzalutamide-resistant conditions and could provide an effective strategy for patients at risk of Enzalutamide resistance and/or for patients who failed Enzalutamide treatment.

The seminal discovery of the role of the androgen receptor (AR) in prostate tumor growth and progression nominated androgen-deprivation therapy (ADT) as the current mainstay for the treatment of advanced prostate cancer (PCa)[1–3]. Despite an initial response to ADT in ~80% of patients[4], relapse and progression to castration-resistant prostate cancer (CRPC) are often inevitable. Interestingly, CRPC tumors retain a high level of AR signaling[5], thus allowing CRPC

patients to benefit from more potent second-generation AR targeting by androgen receptor-signaling inhibitors (ARSIs)[6–11], with Enzalutamide (Enza) as one of the most commonly used ARSIs[9].

Enzalutamide is an AR signaling inhibitor that blocks binding of androgen to AR in addition to inhibiting AR nuclear translocation and its binding to DNA[12]. Administration of Enzalutamide has been shown to improve patient survival overall, yet response to Enzalutamide is

[1]Department of Health Informatics, Rutgers School of Health Professions, Newark, NJ 07107, USA. [2]Department of Urology, Northwestern University Feinberg School of Medicine, Chicago, IL 60611, USA. [3]Rutgers Biomedical and Health Sciences, Rutgers School of Graduate Studies, Newark, NJ 07039, USA. [4]Department of Electrical and Computer Engineering, Rutgers School of Engineering, New Brunswick, NJ 08854, USA. [5]Rutgers Cancer Institute of New Jersey, New Brunswick, NJ 08901, USA. [6]Department of Urology, Yale School of Medicine, New Heaven, CT 06510, USA. [7]Robert H. Lurie Comprehensive Cancer Center, Chicago, IL 60611, USA. [8]These authors contributed equally: Sukanya Panja, Mihai Ioan Truica. [9]These authors jointly supervised this work: Vishal Kothari, Sarki A. Abdulkadir, Antonina Mitrofanova. ✉e-mail: vkothari@pliantrx.com; sarki.abdulkadir@northwestern.edu; amitrofa@shp.rutgers.edu

heterogeneous with nearly half of CRPC patients either not responding and/or developing resistance to Enzalutamide within 8 to 18 months[12–17]. Thus, prioritization of patients based on their risk of developing resistance to Enzalutamide could provide an effective avenue for a personalized line of treatment and potential improvement in overall PCa management.

Recently, several groups have tackled mechanisms involved in response to Enzalutamide, including investigation of the chromatin remodeling protein *CHD1*[18], TGF-β signaling[19], the Persist gene signature[20] (which contains a cohort of genes involved in cell proliferation, cell cycle regulation, stemness, chromatin remodeling and reorganization, and DNA repair), Wnt/β-catenin pathway[21], stemness programs[12] etc. These discoveries provide substantial advances in uncovering mechanisms involved in Enzalutamide response, yet their clinical utility and potential therapeutic intervention remain to be further investigated.

Since MYC (i.e., c-MYC) has been known to play a central role in PCa development and metastasis[3,22–28], is overexpressed in 60% of CRPC patients[27,29], has been shown to be associated with general ARSI response[27], and could potentially be therapeutically targeted[30,31], we sought to investigate MYC mechanisms for their role in response to Enzalutamide.

In this work, we developed TR-2-PATH algorithm to reconstruct a CRPC-specific mechanism-centric regulatory network using the Stand Up to Cancer (SU2C) East Coast cohort[32,33], which identified a network of transcriptional regulatory programs (comprised of transcriptional regulators and their target genes) upstream of the MYC pathway. Querying this network with signatures of favorable and poor Enzalutamide response nominated the NME2 transcriptional regulatory program as the most significant upstream regulatory partner of MYC in CRPC Enzalutamide-associated conditions. We demonstrated consistent association of MYC pathway and NME2 transcriptional regulatory activities across multiple CRPC patient cohorts and showed that increased activity levels of MYC pathway and NME2 transcriptional regulatory program could be utilized as markers to identify primary resistance to Enzalutamide, independent of clinical and molecular variables. Further, we evaluated MYC and NME2 partnership in response to Abiraterone (a second-generation AR signaling inhibitor of androgen biosynthesis)[8,33,34] and did not observe association to the risk of resistance to Abiraterone. Functional studies further demonstrated that therapeutic targeting of MYC using the small-molecule MYC inhibitor MYCi975 and concurrent NME2 knock-down could reverse Enzalutamide resistance. Thus, we propose that MYC-associated mechanisms could be utilized to identify CRPC patients

that are at risk of developing primary resistance to Enzalutamide and that therapeutic targeting of these mechanisms could provide an effective strategy for patients at risk of Enzalutamide resistance and/or for patients who failed Enzalutamide treatment.

## Results

### Increased MYC activity is associated with the risk of Enzalutamide resistance in CRPC patients

We observed that the levels of MYC are increased in Enzalutamide-resistant conditions (treatment with 20 μM Enzalutamide for up to 3 months, EnzaRes) compared to control (DMSO) conditions in LNCaP (a cell line derived from prostate cancer metastasis to lymph-node[35], commonly used to study Enzalutamide resistance) and C42B (LNCaP metastatic CRPC derivative) cell lines (Fig. 1a, b, one-tailed Welch t-test *p* value = 0.002 and *p* value = 0.0018 for LNCaP and C42B cells, respectively, see Methods), which were accompanied by increased levels of AR (Supplementary Fig. 1a, b, one-tailed Welch t-test *p* value = 0.011 and *p* value = 0.001 for LNCaP and C42B cells, respectively, see Methods).

To evaluate if this observation could be translated to human patients, we tested if high MYC pathway activity was characteristic of CRPC patients at risk of developing resistance to Enzalutamide. For this, we utilized RNA-seq profiles of CRPC patients from ref. 33, specifically selecting samples from CRPC patients that did not receive any ARSI treatment prior to sample collection (i.e., ARSI-naïve). We further selected patients that after biopsy (i.e., sample collection), were treated with Enzalutamide and monitored for Enzalutamide-associated disease progression (see Methods, Supplementary Data 1, *n* = 22). We subjected this patient cohort (referred to as Enzalutamide-specific Abida et al. cohort hereafter) to single-sample Gene Set Enrichment Analysis (GSEA)[36] to estimate activity levels of MYC pathway (i.e., HALLMARK_MYC_TARGETS_V2 from Hallmark collection in MSigDB 3.0, n = 58) in each patient, that were then subjected to unsupervised clustering (see Methods), separating patients with high MYC pathway activity (Fig. 1c, yellow, *n* = 15) and normal/low MYC pathway activity (Fig. 1c, blue, *n* = 7). We then compared Enzalutamide-associated disease progression (which was defined in ref. 33 as the time on Enzalutamide without being subjected to another agent, such as taxane) between these groups using Kaplan-Meier survival analysis[37] and Cox proportional hazards modeling[38], which demonstrated a significant difference between patients from high MYC and normal/low MYC pathway activity groups (Fig. 1c, log-rank *p* value = 0.012, adjusted HR (hazard ratio) = 4.39, CI (confidence interval): 1.2–15.97, see Methods), indicating that increased MYC pathway activity is characteristic for

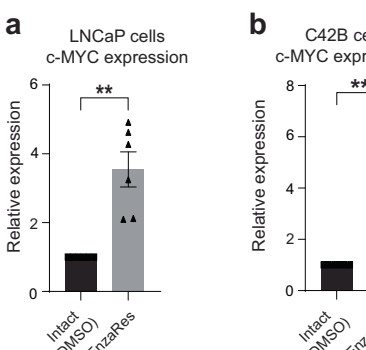
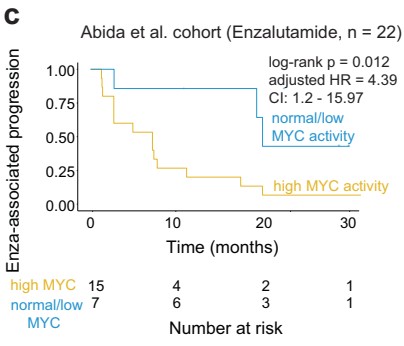
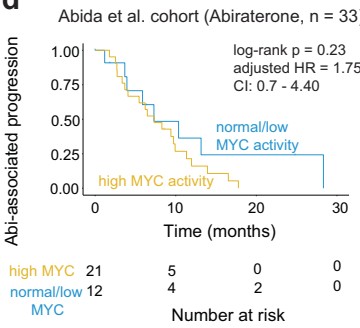

**Fig. 1 | MYC pathway activity is specific for predicting response to Enzalutamide in CRPC patients.** c-MYC expression in Intact (DMSO treated) and Enzalutamide-resistant (EnzaRes) (**a**) LNCaP and (**b**) C42B cells as shown using qRT-PCR. *P* values were estimated using a one-tailed Welch t-test. Data are presented as mean values ± SEM from *n* = 6 independent biological replicates. **\*\****p* value ≤ 0.01. Source data are provided as a Source Data file. Kaplan-Meier survival analysis

comparing CRPC patients that received (**c**) Enzalutamide or (**d**) Abiraterone after sample collection from the Abida et al. cohort with high (yellow) and normal/low (blue) MYC pathway activity levels. Log-rank *p* value, adjusted HR (hazard ratio), and CI (confidence interval) are indicated. Source data are provided as a Source Data file.

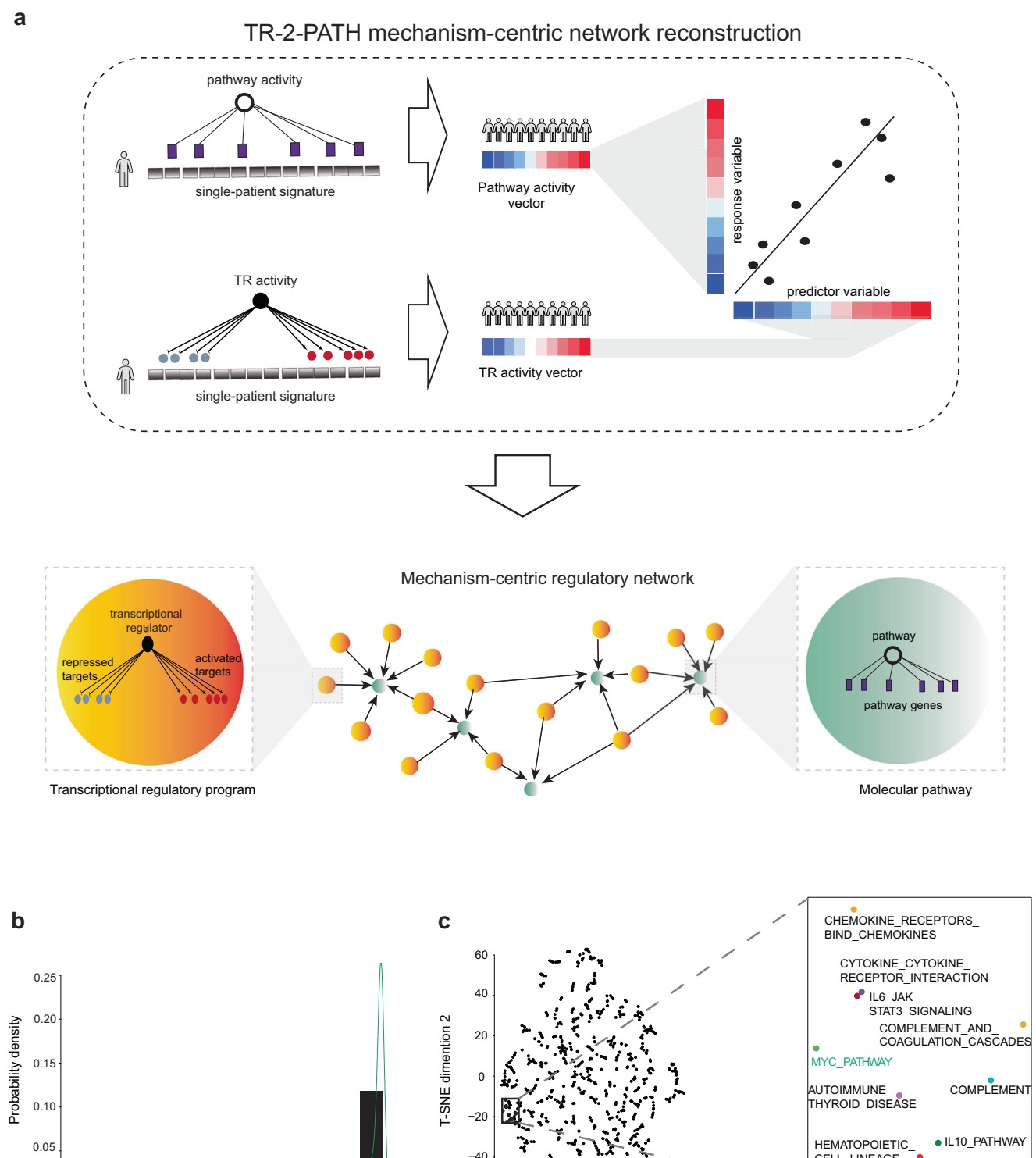

**a** TR-2-PATH mechanism-centric network reconstruction

**b**

**c**

patients with a higher risk of resistance to Enzalutamide. The patient group with high MYC pathway activity also demonstrated increased levels of AR expression and activity (estimated as in ref. 39, see Methods) (Supplementary Fig. 1c, d, one-tailed Welch t-test $p$ value = 0.002 and $p$ value = 0.01, for AR expression and AR activity, respectively) and significant correlation was observed between MYC pathway activity and AR expression/activity levels in the Enzalutamide-specific

Abida et al. cohort[33] ($n$ = 22), as described above (Spearman correlation rho = 0.482, $p$ value = 0.024; and Spearman correlation rho = 0.484, $p$ value = 0.023, for AR expression and AR activity, respectively). To ensure that the correlation between MYC pathway and AR transcriptional regulatory program is not due to their compositional similarity, we evaluated overlap between MYC pathway genes ($n$ = 58, Supplementary Data 2) and AR transcriptional regulon ($n$ = 231,

**Fig. 2 | Reconstruction of a mechanism-centric regulatory network for CRPC patients. a** Schematic representation of the TR-2-PATH workflow. (First row) Single-patient pathway enrichment analysis and single-patient transcriptional regulatory analysis identifies pathway activity vector and transcriptional regulatory activity vector respectively, pairs of which are then subjected to linear regression analysis to reconstruct a mechanism-centric regulatory network. (Second row) In the network, transcriptional regulatory programs are represented as orange nodes and molecular pathways as green nodes. An edge (black arrow) illustrates that a significant relationship was defined between a transcriptional regulatory program and molecular pathway. **b** Distribution of edge weights across the network, as defined by the bootstrap analysis. The x-axis corresponds to the edge weight and the y-axis to its frequency (probability). **c** (Left) t-SNE clustering of molecular pathways (dots), based on the weights of their incoming edges. (Right) Pathways around MYC are shown as a zoom-in and MYC pathway is shown in green. Source data are provided as a Source Data file.

Supplementary Data 2), which demonstrated negligible similarity between these two mechanisms (common genes = 2, Jaccard similarity[40] index = 0.007 out of 1).

Interestingly, in Abiraterone-specific Abida et al. cohort[33], we did not observe a correlation of MYC pathway activity with Abiraterone response. In particular, we utilized RNA-seq profiles of CRPC patients from Abida et al. cohort[33], selecting patients that did not receive any ARSI treatment prior to sample collection and then sub-selecting patients that after biopsy (i.e., sample collection) were treated with Abiraterone and monitored for Abiraterone-associated disease progression (which was defined in ref. 33 as the time on Abiraterone without being subjected to another agent, see Methods, Supplementary Data 1, $n = 33$, referred to as Abiraterone-specific Abida et al. cohort). We identified no significant difference in Abiraterone-associated disease progression between patients from high MYC and normal/low MYC groups (Fig. 1d, log-rank $p$ value = 0.23, adjusted HR = 1.75, CI: 0.7–4.40, see Methods) in the Abiraterone-specific Abida et al. cohort.

Taken together, these analyses demonstrate that increased activity of MYC pathway could potentially serve as a marker to stratify patients for their risk of developing resistance to Enzalutamide and would benefit from validation in larger patient cohorts for future clinical use.

## Defining MYC upstream regulatory programs through mechanism-centric network analysis

To elucidate MYC regulation implicated in Enzalutamide resistance and define potential additional axes for salvage therapeutics, we investigated transcriptional regulatory mechanisms upstream of MYC pathway that might affect MYC while also governing Enzalutamide resistance. For this, we developed "TR-2-PATH" algorithm to reconstruct a CRPC-specific mechanism-centric regulatory network, which connects molecular pathways (Fig. 2a, green nodes) with potential upstream transcriptional regulatory (TR) programs (Fig. 2a, orange nodes). Nodes in this mechanism-centric network do not correspond to individual genes or alterations, but rather represent mechanisms: such as transcriptional regulatory programs or molecular pathways, promising to uncover complexity underlying therapeutic resistance and identify more effective biomarkers and optimized therapeutic interventions (Fig. 2a). Our objective was to reconstruct a mechanism-centric regulatory network that would capture a wide-array of CRPC-specific phenotypes, constituting a valuable resource for the community and allowing effective utilization across different CRPC-related questions (e.g., primary and secondary therapeutic resistance, metastases to different sites etc.) For this, we utilized RNA-seq profiles from CRPC patients in the Stand Up to Cancer (SU2C) East Coast cohort[32,33], excluding repeated samples and samples from ref. 33 (see Methods, Supplementary Data 1, $n = 153$). The selected SU2C East Coast cohort was well-suited for CRPC mechanism-centric network reconstruction as it (i) constitutes one of the largest-to-date cohorts of CRPC patients, essential for statistical learning/inference; (ii) is characterized by wide-ranged age (59.2 ± 8.38 years) and prostate specific antigen (PSA) levels (234.5 ± 1574.4 ng/ml); (iii) includes different metastatic sites, including bone ($n = 39$), liver ($n = 26$), lymph-node ($n = 57$), prostate ($n = 4$), lung ($n = 2$), adrenal ($n = 1$), other soft tissue ($n = 19$), etc.; and (iv) represents different stages of therapeutic intervention,

including samples from patients previously exposed to ARSIs (including Enzalutamide and Abiraterone, $n = 67$), ARSI-naïve at the time of sample collection ($n = 75$), currently on treatment ($n = 4$), etc.; all together capturing a wide range of clinical variables necessary for accurate statistical inference.

For network reconstruction, to evaluate relationships between molecular pathways and their upstream transcriptional regulatory programs, we first needed to estimate pathway activity levels and transcriptional regulator activity levels in each sample in the SU2C East Coast cohort (Fig. 2a). To estimate pathway activity levels, we performed single-sample GSEA[36] on the scaled SU2C East Coast cohort, so that each sample ($n = 153$) was used as a reference signature and each molecular pathway ($n = 883$, from KEGG[41], BioCarta[42], Reactome[43], and Hallmark[44] collections) was used as a query (see Methods). To estimate activity of TRs in the SU2C East Coast cohort, we performed VIPER analysis[45] on the scaled SU2C East Coast cohort (as above) utilizing each sample ($n = 153$) as a reference and transcriptional regulatory programs ($n = 2678$) from a prostate cancer specific interactome[39] as a query (see Methods). For each molecular pathway, its activity level (defined as Normalized Enrichment Scores, NES) across all patients in the cohort then defined a "pathway activity vector" (Fig. 2a, top, see Methods). Similarly, for each transcriptional regulatory program, its activity level across all patients in the cohort defined a "TR activity vector" (Fig. 2a, bottom, see Methods). All pairs of TR-pathway activity vectors were then subjected to linear regression analysis[46] (see Methods), where "TR activity vector" was utilized as a predictor (independent) variable and "pathway activity vector" was used as a response (dependent) variable, with an objective to identify TR programs whose changes could potentially explain changes in the activity of molecular pathways in CRPC-specific manner. Significant relationships, corrected for multiple hypotheses testing (see Methods), between TRs and pathways (both positive and negative) were then considered for network reconstruction (Fig. 2a).

To ensure that the network is robust to experimental and sampling noise, we enhanced our network reconstruction with bootstrap analysis. For this, patients from the SU2C East Coast cohort ($n = 153$) were sampled with replacement (see Methods, $k = 100$) and each bootstrap was subjected to TR-2-PATH network reconstruction. A comparison of edge distributions across the 100 bootstrapped networks showed their similarity (Supplementary Fig. 2), indicating the method's overall reproducibility. A total of 100 bootstrapped networks were then used to assign "weight" to each edge in the network reconstructed from the whole dataset ($n = 153$), reflecting the number of times an edge appears (and thus could be recovered) across the bootstrapped networks (see Methods, Supplementary Data 3, Fig. 2b). Unsupervised t-distributed stochastic neighbor embedding clustering[47] (t-SNE) was utilized to cluster molecular pathways based on weights of their incoming edges, demonstrating co-clustering of MYC pathway with Chemokine[48], Cytokine[49], IL-6[50], JAK STAT 3 signaling[50], and IgA[51] pathways (see Methods, Fig. 2c), and revealing their potential cross-talk in the CRPC setting[52].

## Network mining I: identifying differentially altered sub-networks

The next step was to utilize this network to identify TR programs upstream of MYC pathway that are involved in Enzalutamide response

## Network mining I: identifying differentially altered sub-networks

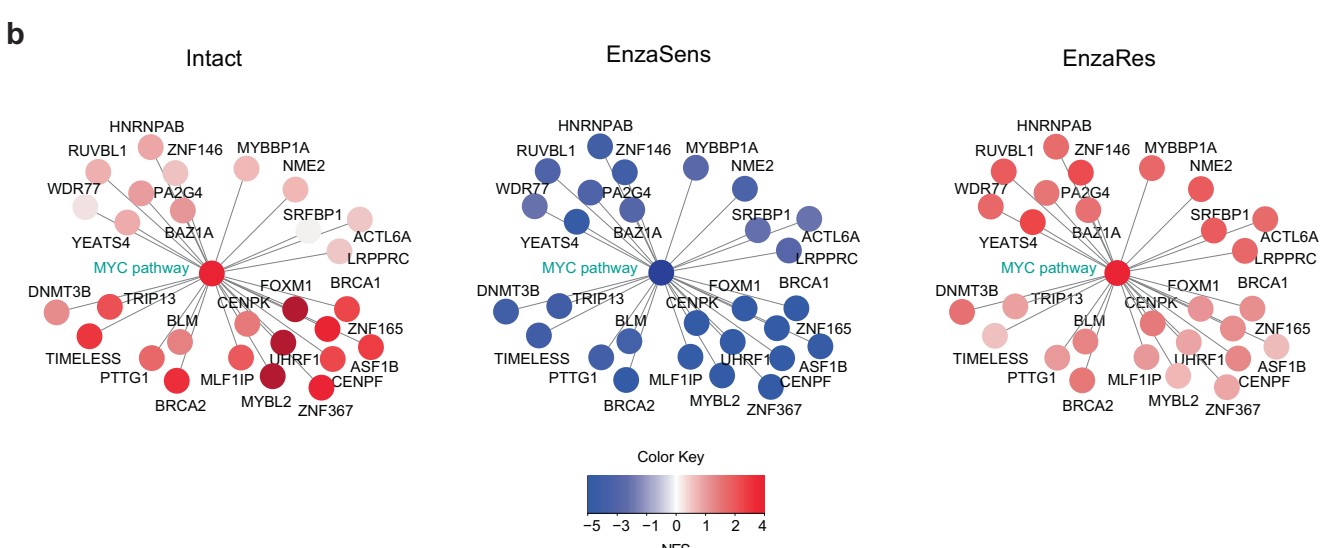

**Fig. 3 | Network mining I: identifying upstream transcriptional regulatory programs that affect MYC pathway and are associated with response to Enzalutamide. a** Schematic representation of the changes in activity levels of molecular pathways and their upstream transcriptional regulatory programs (i.e., sub-networks) as they transition from Intact (treated with DMSO) to Enzalutamide-sensitive (EnzaSens) to Enzalutamide-resistant (EnzaRes) phenotypes. Red depicts up-regulation and blue depicts down-regulation of TR and pathway activities. **b** Identified upstream transcriptional regulatory programs (MYC-centered sub-network) associated with Enzalutamide treatment affecting MYC pathway, depicted across Intact, EnzaSens, and EnzaRes phenotypes. Red depicts up-regulation and blue depicts down-regulation of TR and pathway activities. Source data are provided as a Source Data file.

and resistance. To achieve this, we aimed to identify parts (sub-networks) of the mechanism-centric network that significantly change (alter) between phenotypes of interest, in our case—phenotypes that describe response to Enzalutamide (Fig. 3a). To accurately capture response and resistance to Enzalutamide in a controlled setting, we utilized gene expression profiles from ref. 53 (see Methods, Supplementary Data 1, $n = 12$), which is based on the LNCaP experimental system that has been widely used to study Enzalutamide-resistance previously[9,54-56]. These profiles included (i) LNCaP parental intact samples subjected to DMSO (phenotype 1 - Intact, Fig. 3a); (ii) LNCaP samples treated for 48 h with Enzalutamide, where their survival and proliferation were sensitive to Enzalutamide (phenotype 2 – EnzaSens, Fig. 3a); and (iii) LNCaP samples treated with Enzalutamide for 6 months, where their survival and proliferation did not depend on Enzalutamide (phenotype 3 – EnzaRes, Fig. 3a).

We hypothesized that regulatory programs that are active in the intact state, then are repressed by Enzalutamide treatment (EnzaSens phenotype) and further re-activated as Enzalutamide resistance develops (EnzaRes phenotype) would be effective candidates to uncover mechanisms that govern Enzalutamide-resistance (Fig. 3a). Such network mining (using pairwise phenotype comparison, see Methods, Supplementary Fig. 3a, b, Supplementary Data 4A–F)

identified TR mechanisms ($n = 28$) upstream of the MYC pathway, which constitutes of MYC-centric TR sub-network with a significant "active->repressed->reactivated" activity changes across the Enzalutamide-related phenotypes (Fig. 3b).

### Network mining II: Prioritization of upstream regulatory programs

The next essential step was to prioritize the identified transcriptional regulatory programs upstream of a pathway of interest (e.g., MYC pathway) for experimental validation and potential salvage therapeutic targeting. We developed such a prioritization step to overcome several important drawbacks, commonly present in widely utilized statistical analyses. First of all, our method considers potential multi-collinearity among input variables (TRs), which is often naturally present in biological systems, yet can substantially obstruct statistical learning. In fact, variance inflation factor (VIF) analysis[57] of the 28 TR programs identified upstream of MYC demonstrated significant multi-collinearity (all TRs had VIF >10, a multi-collinearity threshold suggested in refs. 58,59) (see Methods, Supplementary Fig. 4) and thus requires special methods to avoid information loss or model mis-interpretation. Commonly utilized methods for handling multi-collinearity (e.g., regularization techniques), often keep one of the

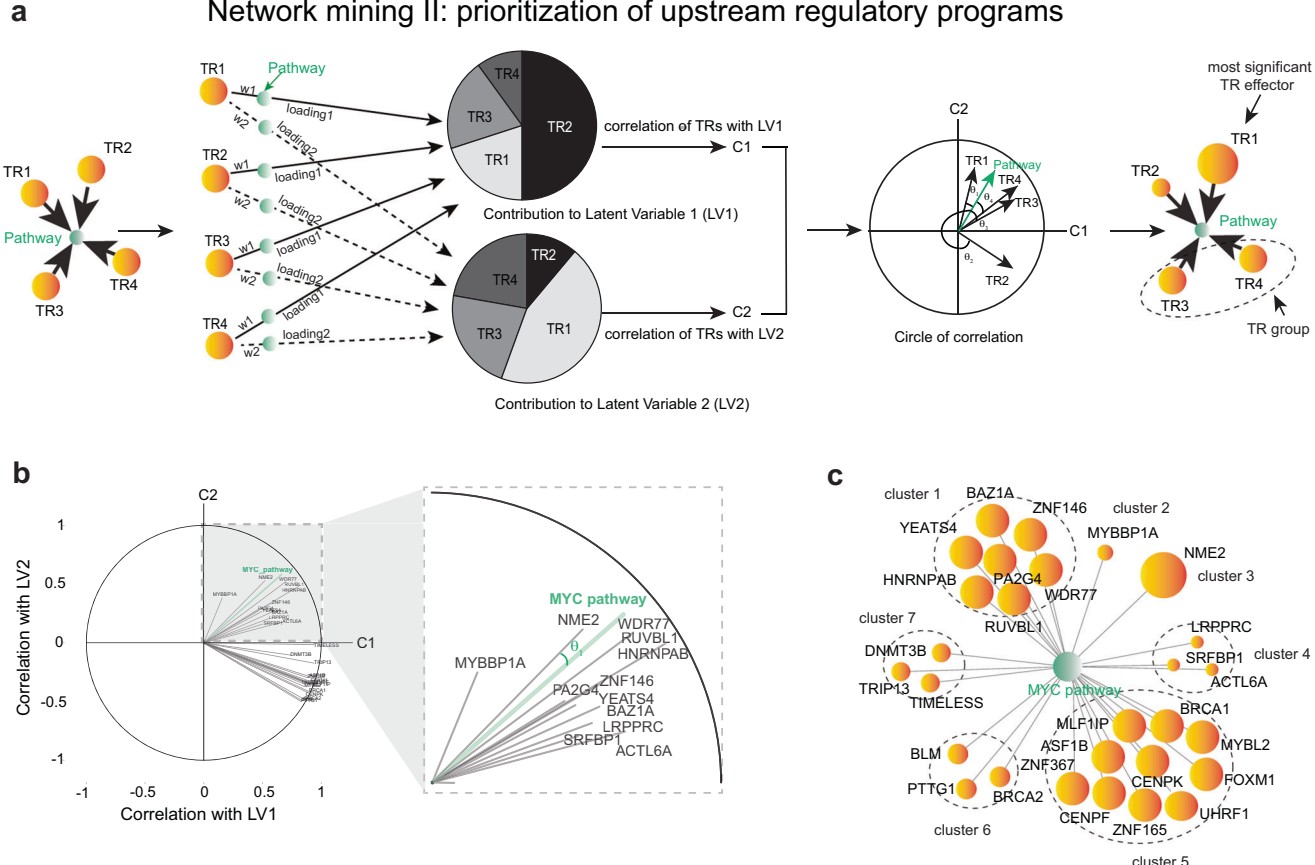

**Fig. 4 | Network mining II: NME2 is prioritized as TR with the most significant effect on MYC pathway. a** Schematic representation of the PLS-inspired approach to prioritize TR programs, based on their effect on a molecular pathway of interest. (Left) TR activity vectors are utilized as inputs, which are then regressed on a pathway to identify non-collinear latent variables (pie charts), which include a linear combination of TR programs, based on their effect on the pathway (slices in each pie). (Middle) These latent variables are utilized to build a circle of correlation, which depicts the relationship between each latent variable and each TR and pathway. (Right) Effect scores are defined to group and prioritize TRs, based on their effect on a pathway. **b** A circle of correlation is utilized to determine the degree of closeness between TR programs and the MYC pathway, based on their effect on each latent variable. **c** Grouping and prioritization of the MYC upstream TR programs. Circle sizes correspond to the TR effect scores. NME2 is determined to have the most significant effect on MYC pathway. Source data are provided as a Source Data file.

collinear variables in the model and eliminate others at random, thus limiting biological interpretability and translatability of the model. Our technique keeps all input variables intact, instead identifying their potential groups (based on their effect on the pathway of interest) and preventing information loss. Furthermore, our prioritization method not only tests for direct regulatory relationships but also considers that a meaningful regulatory relationship can exist between entities that do not necessarily have direct (but rather indirect) interactions, which are widely present in biological systems, potentially including relationships between TRs and biological pathways.

To overcome these limitations, we developed a prioritization step, inspired by the Partial Least Squares (PLS) approach[60], which has been mostly utilized in social sciences[61–64] (sometimes referred to as a "supervised PCA") but has not been used for network-based mining in oncology to date. Briefly, to identify TR groups and prioritize the effect of the TR programs on the MYC pathway, our approach considers TR activity vectors as predictor (input) variables and utilizes MYC pathway activity vector as a response (output) variable (see Methods, Fig. 4a, left). It then regresses TR activity vectors on the MYC pathway vector so that a linear combination of TRs defines a latent variable (see Methods, Fig. 4a, left). This latent variable is then "subtracted" from the TR activity vectors, leaving the residuals to be utilized for defining the next latent variable (see Methods). Identified latent variables do not express collinearity or multi-collinearity and are utilized as axes to

build a "circle of correlation" (see Methods, Fig. 4a, middle). Such a circle of correlation depicts the association of TR programs and the MYC pathway (defined as arrows on the circle of correlation, see Methods) to each latent variable. We then developed a method that utilized unsupervised hierarchical clustering on the degrees of closeness (angles) between TR and pathway arrows (see Methods) so that TRs in high proximity to one another (thus having similar effects on latent variables) are grouped as they express simultaneous effect on the MYC pathway (see Methods, Fig. 4a, right). Such TR groups/clusters (which also include groups with one TR) are then "prioritized" based on their effect on the MYC pathway activity (see Methods, Fig. 4a, right) using effect scores, which are defined as a combination of (i) degree of closeness between a TR group/cluster and the MYC pathway on the circle of correlation (i.e., angle between their arrows), which reflects effect of each TR group activity changes on MYC pathway; (ii) association (i.e., Pearson correlation) between a TR group/cluster and each evaluated latent variable, which reflects contribution of each TR group to each latent variable; and (iii) edge weight between TR group/cluster and the MYC pathway from the TR-2-PATH mechanism-centric network reconstruction step, which reflects robustness of their regulatory relationship (see Methods, Fig. 4b).

This approach identified 7 TR groups/clusters, based on their effect on the MYC pathway activity (two of the clusters had single TRs, Fig. 4c): group/cluster 1 (HNRNPAB, YEATS4, BAZ1A, ZNF146, WDR77,

RUVBL1, and PA2G4), group/cluster 2 (MYBBP1A), group/cluster 3 (NME2), group/cluster 4 (ACTL6A, LRPPRC and SRFBP1), group/cluster 5 (FOXM1, MYBL2, BRCA1, MLF1IP, ASF1B, ZNF367, CENPF, ZNF165, CENPK, and UHRF1), group/cluster 6 (BRCA2, PTTG1, and BLM), and group/cluster 7 (TIMELESS, TRIP13, and DNMT3B) (Fig. 4c). Each group/cluster was then assigned an effect score, with group/cluster 3 (NME2)[65] having the highest effect (score) on the MYC pathway (Fig. 4c, Supplementary Fig. 5a, b, Supplementary Data 5). While our analysis nominated NME2 transcriptional regulatory program to have the highest effect on MYC pathway in Enzalutamide-associated CRPC context, it has also been previously shown to bind to the *MYC* promoter region and upregulate *MYC* transcription[44,66], suggesting that further investigation of this relationship may uncover aspects of the transcriptional regulatory mechanisms governing the MYC pathway which could potentially provide an additional axis for therapeutic targeting for CRPC patients.

## Validation in clinical cohorts

We next sought to confirm and evaluate if activation of NME2 TR program and MYC pathway are present in patients at risk of resistance to Enzalutamide and if they can be used as markers to risk-stratify patients prior to Enzalutamide administration. First, to evaluate if activity of the MYC pathway and NME2 TR program are high in treatment-naïve patients, who are at risk of developing resistance to Enzalutamide, we have evaluated single-cell profiles from two sequential samples from a CRPC patient (01115655) that eventually failed Enzalutamide: neoadjuvant sample (prior to Enzalutamide treatment, Fig. 5a) and adjuvant sample (after developing resistance to Enzalutamide, i.e., EnzaRes, Fig. 5b) from ref. 19 (see Methods, Supplementary Data 1). After subjecting single-cell transcriptomic data to unsupervised uniform manifold approximation and projection (UMAP) clustering[67] (see Methods), to identify adenocarcinoma cells among the cell populations we assessed activity levels of AR, alongside expression of CK8 and CD45 (Supplementary Fig. 6a–d). Following adenocarcinoma identification, we evaluated activity levels of the MYC pathway and NME2 TR program in both neoadjuvant and adjuvant samples. Our analysis indicated significantly higher levels of both NME2 TR activity and MYC pathway activity in adenocarcinoma cells, compared to other cells in Enzalutamide-naïve (neoadjuvant, Fig. 5a, one-tailed Welch t-test $p$ value < 2.26E-16 for NME2 and $p$ value = 1.74E-7 for MYC, see Methods) and Enzalutamide-resistant, i.e., EnzaRes (adjuvant, Fig. 5b, one-tailed Welch t-test $p$ value < 2.26E-16 for NME2 and $p$ value < 2.26E-16 for MYC, see Methods) samples, indicating that (i) both high-MYC pathway and high-NME2 TR activity levels were present prior to treatment in a patient who was at risk of developing subsequent resistance to Enzalutamide, nominating them as markers to identify patients at potential risk of Enzalutamide resistance; and (ii) both high-MYC pathway and high-NME2 TR activity levels were also observed after resistance to Enzalutamide developed (similar to our observation in LNCaP and C42B cell lines, Fig. 1a, b, Supplementary Fig. 5a, b), cautiously nominating a MYC-centered salvage therapeutic line for patients that fail Enzalutamide.

Given increased activity levels of both NME2 TR and MYC pathway in a single-cell sample from a treatment-naïve patient that later developed resistance to Enzalutamide, we sought to confirm their collective ability to predict CRPC patients at the treatment-naïve stage for risk of developing primary resistance to Enzalutamide using the Abida et al.[33] cohort, which was also utilized in Fig. 1c (see Methods, Supplementary Data 1, $n = 22$). As previously described, we used Enzalutamide-specific Abida et al. cohort[33] (i.e., ARSI-naïve CRPC patients that were later subjected to Enzalutamide and monitored for Enzalutamide-associated disease progression) to estimate NME2 TR and MYC pathway activity levels in each patient (see Methods). The NME2 TR and MYC pathway demonstrated

concordance of their activity levels while maintaining largely distinct regulatory programs (overlap between NME2 transcriptional regulon $n = 412$ and MYC pathway $n = 58$ is equal to 7 genes, Jaccard similarity index = 0.01 out of 1, Supplementary Data 6) (Fig. 5c, left top, Pearson $r = 0.8$, $p$ value = 5.2E-6, see Methods). The association between NME2 transcriptional activity and MYC pathway activity was also confirmed in refs. 12 ($n = 24$), 68 (CRPC samples $n = 34$) and 69 ($n = 157$) CRPC patient cohorts (Alumkal et al. Pearson $r = 0.74$, $p$ value = 3.53E-5; Beltran et al. Pearson $r = 0.86$, $p$ value = 5.13 E-11; Kumar et al. Pearson $r = 0.65$, $p$ value < 2.2E-16, Supplementary Fig. 7, Supplementary Data 1, see Methods).

Next, we subjected the activity levels of the NME2 TR and MYC pathway in Enzalutamide-specific Abida et al.[33] cohort to unsupervised clustering (see Methods) that identified (i) patients with both high levels of NME2 transcriptional activity and MYC pathway activity (Fig. 5c, left bottom, n = 13, yellow group) and (ii) patients with at least one low/normal NME2 transcriptional activity and/or MYC pathway activity, categorized as "others" (Fig. 5c, left bottom, $n = 9$, blue group). A comparison of these groups using Kaplan-Meier survival analysis[37] and Cox proportional hazards model analysis[38], demonstrated a significant difference in their Enzalutamide-associated disease progression (Fig. 5c, right, log-rank $p$ value = 0.0035, adjusted HR = 5.28, CI = 1.58–18.38, see Methods).

To extend our validation studies to an additional patient cohort, we utilized SU2C West Coast cohort[70,71] (see Methods, Supplementary Data 1, $n = 83$) which comprises of CRPC patients that were subjected to Enzalutamide and/or Abiraterone (~67% of patients in this cohort were pre- or post-treated with Enzalutamide, either alone or in combination with Abiraterone) either before or after biopsy (i.e., before or after sample collection) and were subsequently monitored for treatment-associated disease progression (which was defined in refs. 70,71 as increase in PSA level (minimum 2 ng/mL) that has risen at least twice, in an interval of at least one week or soft tissue progression (nodal and visceral) based on RECIST v1.1). For each of these CRPC patients, we first estimated their NME2 TR and MYC pathway activity levels (see Methods) followed by evaluating association between NME2 TR and MYC pathway activity. Our analysis demonstrated concordance between NME2 TR and MYC pathway activity (Fig. 5d, left top, Pearson $r = 0.82$, $p$ value < 2.2E-16, see Methods). Next, we subjected the activity levels of the NME2 TR and MYC pathway to unsupervised clustering (see Methods) that identified (i) patients with both high NME2 TR and MYC pathway activity levels (Fig. 5d, left bottom, $n = 40$, yellow group) and (ii) patients with at least one low/normal NME2 TR and/or MYC activity levels, categorized as "others" (Fig. 5d, left bottom, $n = 43$, blue group). A comparison of these groups using Kaplan-Meier survival analysis[37] and Cox proportional hazards model analysis[38] demonstrated a significant difference in treatment-associated disease progression (Fig. 5d, right, log-rank $p$ value = 0.026, adjusted HR = 1.90, CI = 1.11–3.24, see Methods), which supports our previous observations.

To evaluate association of NME2 TR and MYC pathway with response to Abiraterone, we analyzed RNA-seq profiles from two Abiraterone-specific cohorts: (i) ARSI-naïve CRPC patients from ref. 33 that were subjected to Abiraterone after sample collection and monitored for Abiraterone-associated disease progression, as in Fig. 1d (see Methods, Supplementary Data 1, $n = 33$) and (ii) PROMOTE[34] cohort, which is comprised of ARSI-naïve CRPC patients, subjected to Abiraterone for 12 weeks after sample collection and then assessed for disease progression (binary outcomes, where disease progression was defined based on the combined score that included serum PSA level, bone and CT imaging and symptom assessments at week 12, see Methods, Supplementary Data 1, $n = 77$). In both cohorts, we estimated the NME2 TR and MYC pathway activity in each sample and subjected them to similar analyses as above. Kaplan-Meier survival analysis[37] and Cox proportional hazards model analysis[38] on the Abida et al.[33] cohort

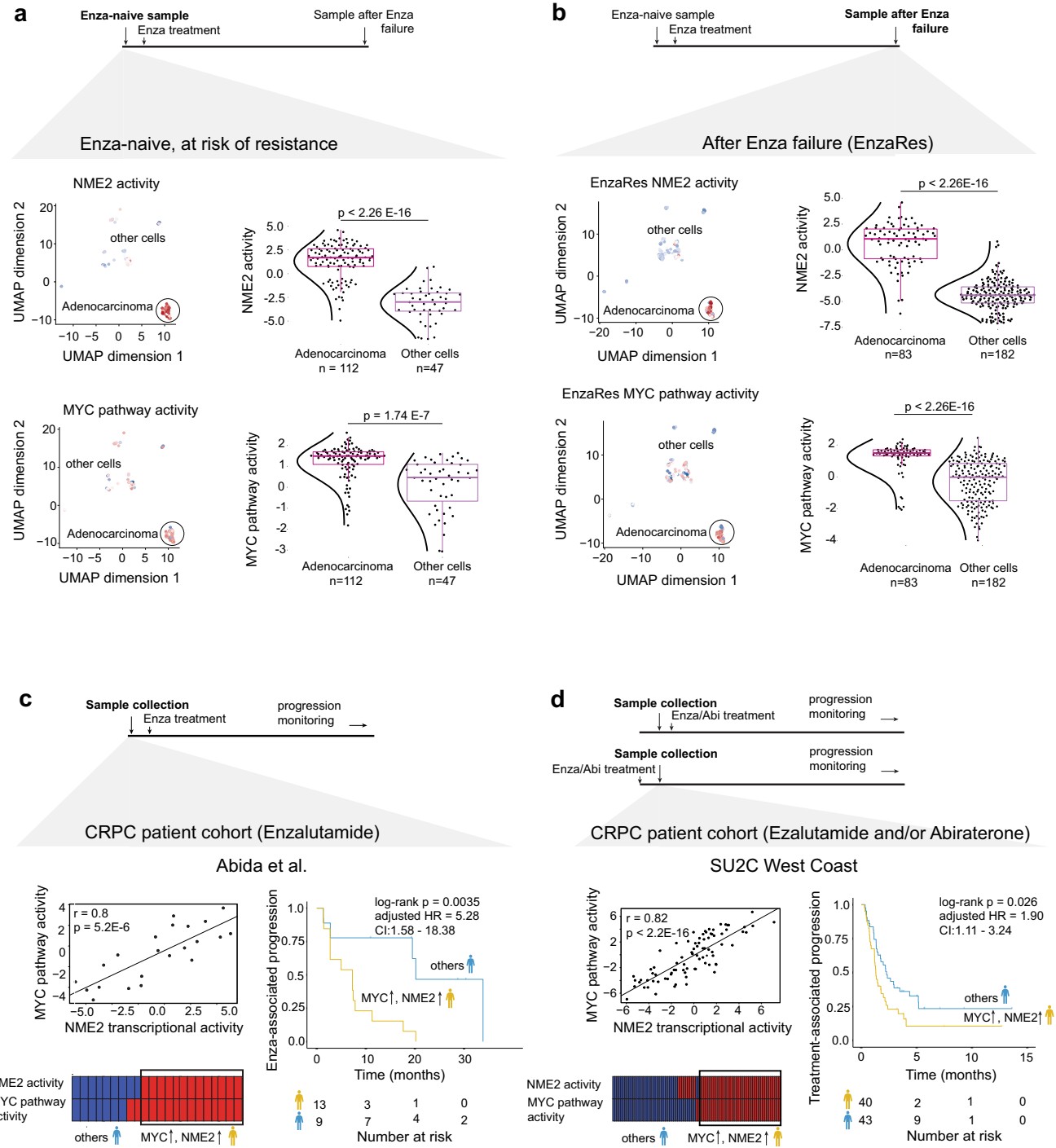

**Fig. 5 | Activities of MYC and NME2 are associated with poor response to Enzalutamide. a**, **b** Comparing NME2 TR and MYC pathway activity between adenocarcinoma cells (dark magenta) and other cells (dark orchid) in neoadjuvant and adjuvant samples of a CRPC patient obtained from He et al. *P* value was esti-mated using a one-tailed Welch t-test. In boxplots, the center line corresponds to the median and the box limits correspond to the first and third quartiles (the 25th and 75th percentiles). The upper and lower whiskers extend to the maximum or minimum value within 1.5 times the interquartile range, respectively. **c** (Left, top) Pearson correlation analysis between NME2 TR and MYC pathway activity in the Abida et al. cohort subjected to adjuvant Enzalutamide. Pearson r and *p* value are indicated. (Left, bottom) Patients with high-MYC and high-NME2 (yellow) and the rest of the patients (blue) were identified. (Right) Kaplan-Meier survival analysis,

comparing high-MYC and high-NME2 group (yellow) to the rest of the patients (blue) in Abida et al. cohort subjected to adjuvant Enzalutamide. Log-rank *p* value, adjusted HR (hazard ratio), and CI (confidence interval) are indicated. **d** (Left, top) Pearson correlation analysis between NME2 TR and MYC pathway activity in SU2C West Coast cohort subjected to Enzalutamide and/or Abirterone either before or after sample collection. Pearson r and *p* value are indicated. (Left, bottom) Patients with high-MYC and high-NME2 (yellow) and the rest of the patients (blue) were identified. (Right) Kaplan-Meier survival analysis, comparing high-MYC and high-NME2 group (yellow) to the rest of the patients (blue) in SU2C West Coast cohort. Log-rank *p* value, adjusted HR (hazard ratio), and CI (confidence interval) are indicated.

## Comparison to markers of overall PCa aggressiveness

**a** Transcriptomic markers

**b** Genomic markers

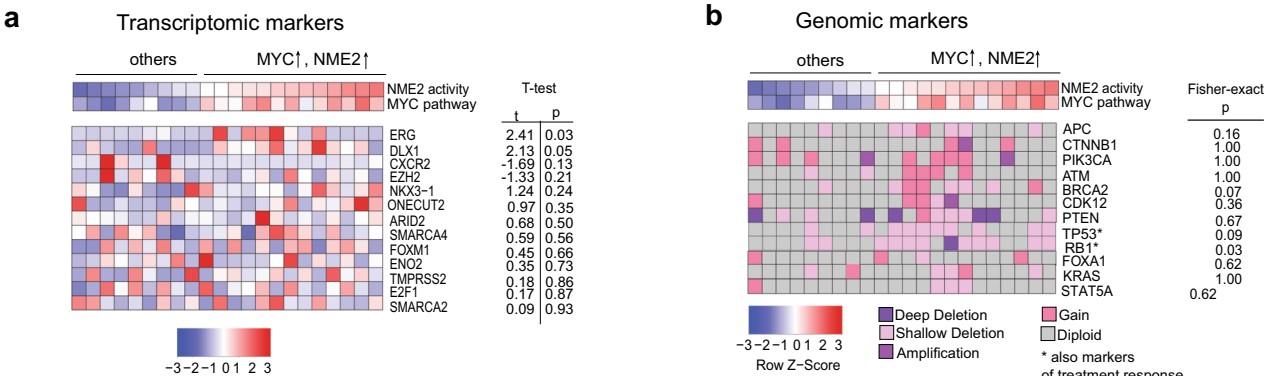

## Comparison to markers of PCa treatment response

**c** Transcriptomic markers of response to ADT and ARSI

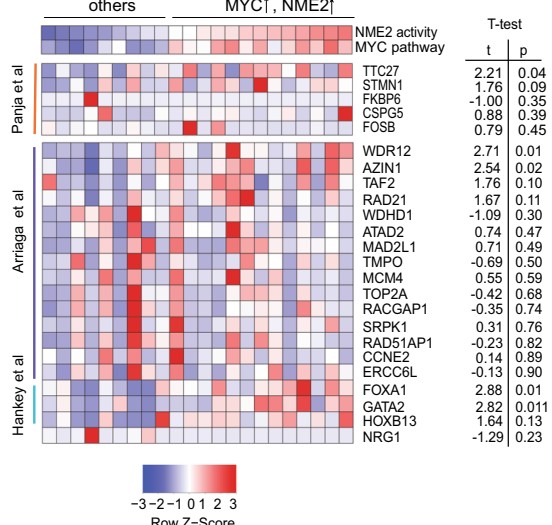

**d** Genomic markers of response to ARSI

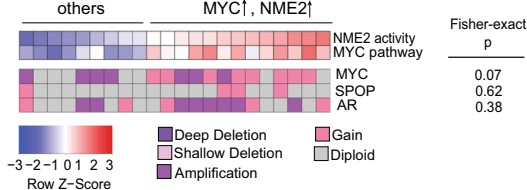

**f** Genomic markers of response to Enza

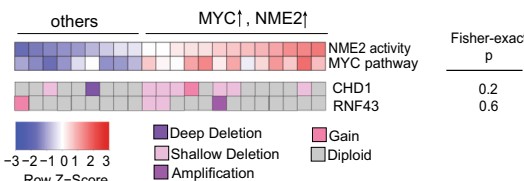

**e** Transcriptomic markers of response to Enza

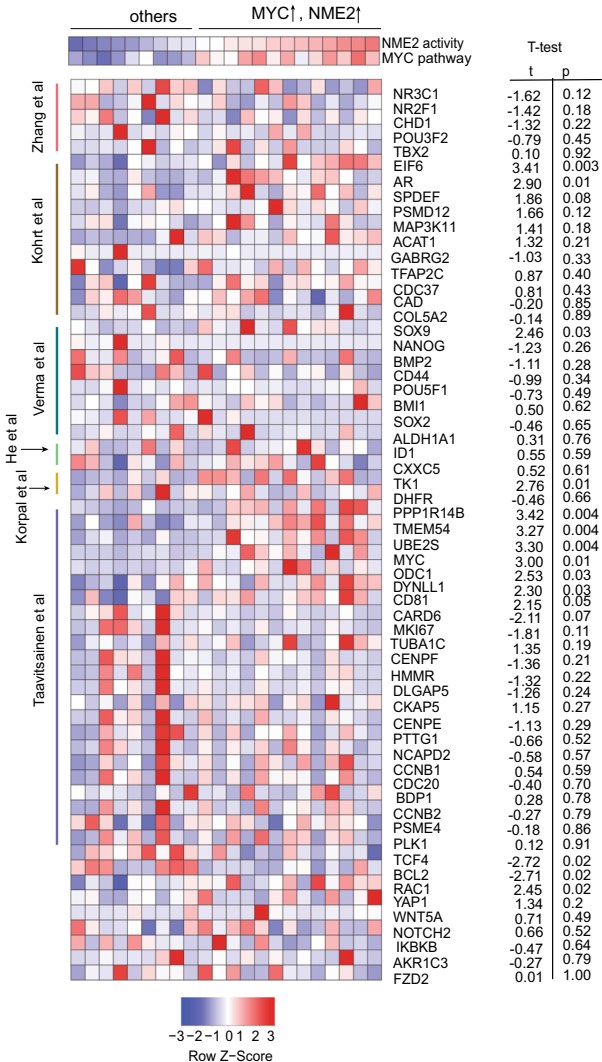

demonstrated no significant difference in Abiraterone-associated disease progression between the two identified patient groups (Supplementary Fig. 8, left, log-rank *p* value = 0.09, adjusted HR = 2.37, CI = 0.92–6.09, see Methods). ROC analysis[72] in the PROMOTE[34] cohort (see Methods) demonstrated that activation of NME2 and MYC did not classify patients based on their binary response to Abiraterone (Supplementary Fig. 8, right, AUROC = 0.58, where AUROC = 0.5 indicates a random classifier). Taken together, these analyses suggest that partnership between NME2 TR and MYC pathway is associated with resistance to Enzalutamide.

**Fig. 6 | Ability of MYC and NME2 to predict Enzalutamide-response outperforms known markers of PCa progression and treatment response.** Comparison of MYC and NME2 ability to predict response to Enzalutamide in Abida et al. cohort to known markers of PCa aggressiveness, including (**a**) transcriptomic and (**b**) genomic markers. Comparison of MYC and NME2 ability to predict response to Enzalutamide in Abida et al. cohort to known markers of response to ADT and ARSIs including (**c**) transcriptomic and (**d**) genomic markers. Comparison of MYC and NME2 ability to predict response to Enzalutamide in Abida et al. cohort to known markers of Enzalutamide-response, including (**e**) transcriptomic and (**f**) genomic markers. Two-tailed Welch t-test was utilized to calculate $p$ values to estimate the difference in expression levels (red corresponds to over-expression and blue to under-expression) between high-MYC and high-NME2 group and the rest of the patients for transcriptomic markers in (**a**, **c**, **e**). Two-tailed Fisher-exact test was utilized to calculate $p$ values to estimate the difference in the frequency/occurrence of any genomic alterations between high-MYC and high-NME2 group and the rest of the patients for genomic markers in (**b**, **d**, **f**).

## Comparison to common markers of PCa aggressiveness and treatment response

We next compared the ability of NME2 TR and MYC pathway to predict Enzalutamide resistance in Abida et al. cohort[33], with the predictive ability of known markers of prostate cancer (i) aggressiveness, (ii) response to first-generation ADT and ARSIs (not specific to any particular drug), and (iii) Enzalutamide-specific response (Fig. 6). Transcriptomic and genomic markers were considered separately.

First, we evaluated the enrichment of transcriptomic markers of PCa aggressiveness[73–86] ($n = 13$, Fig. 6a) in high NME2 and MYC group, out of which *ERG, and DLX1* showed significant differential expression in the high-MYC and high-NME2 group, compared to the rest of the patients, "others" (two-tailed Welch t-test $p$ value < 0.05, Supplementary Data 7A, see Methods). Interestingly, these genes have a direct relationship to MYC: *ERG* fusion (which eventually leads to over-expression of *ERG*) was shown to be correlated to *MYC* expression[87] and *DLX1* is a known transcriptional target of ERG[76,88,89] (Fig. 6a). To assess an independent association of the transcriptomic markers to Enzalutamide resistance (independent of MYC and NME2), we performed a univariable Cox proportional hazards model analysis[38], using Enzalutamide-associated disease progression as the end-point in Enzalutamide-specific Abida et al.[33] cohort, which showed no significant association of these markers to Enzalutamide resistance (Supplementary Data 7A, see Methods). Among genomic markers[21,32,90–92] of PCa aggressiveness ($n = 12$, Fig. 6b, where *TP53* and *RB1* have also been shown to be markers of response to ARSIs[33]), *RB1* with shallow and deep deletion was significantly enriched in patients with high-NME2 and high-MYC pathway activity (Fisher's exact test $p$ value = 0.03) (Fig. 6b, Supplementary Data 7B, see Methods). Interestingly, *RB1* loss has been shown to correlate with *MYC* expression in small-cell lung carcinoma[93], indicating potential cross-talk with the MYC pathway. Independent Cox proportional hazards model analysis[38] identified a shallow and deep deletion of *KRAS* to be significantly associated with response to Enzalutamide (Wald $p$ value = 0.01, Supplementary Data 5B, see Methods), which has also been previously shown to be associated with MYC[27].

Comparison to transcriptomic markers of first-generation ADT and ARSIs ($n = 24$), taken from refs. 27,94–96), demonstrated that five markers (*TTC27, WDR12, AZIN1, FOXA1, and GATA2*) were significantly differentially expressed in the high-MYC and high-NME2 patients (two-tailed Welch t test $p$ value < 0.05, Supplementary Data 5C, Fig. 6c, see Methods). Interestingly, *WDR12* and *AZIN1* are members of the MYC pathway and *TTC27, FOXA1, and GATA2* are MYC transcriptional targets[86]. Furthermore, Cox proportional hazards model analysis[38] indicated that five of the transcriptomic markers (*STMN1, WDR12, AZIN1, MAD2L1, and MCM4*) had a significant association with response to Enzalutamide (Wald $p$ value < 0.05, Supplementary Data 5C, see Methods), yet many of them were borderline significant and did not outperform MYC and NME2 (Supplementary Data 7C). Additionally, genomic markers of first-generation ADT and ARSIs described in refs. 27, 33 ($n = 3$), had no significant enrichment in the high-MYC and high-NME2 group (Fig. 6d, Supplementary Data 7D, see Methods) or independent response to Enzalutamide.

Comparison to transcriptomic markers of Enzalutamide-specific response ($n = 60$), described by refs. 18,97–100,20, in addition to

refs. 101–108) demonstrated that a group of 14 markers (*EIF6, AR, SOX9, TK1, PPP1R14B, TMEM54, UBE2S, MYC, ODC1, DYNLL1, CD81, BCL2, TCF4 and RAC1*) had significant differential expression in the high-MYC and high-NME2 group (two-tailed Welch t-test $p$ value < 0.05, Supplementary Data 7E, Fig. 6e, see Methods). Interestingly, 12 of these (*EIF6, SOX9, TK1, PPP1R14B, TMEM54, UBE2S, MYC, ODC1, DYNLL1, CD81, BCL2, and TCF4*) were transcriptional MYC targets, determined from ChEA transcription factor targets dataset[86]. Another member of this group, *AR*, as we have shown previously (Supplementary Fig. 1c), was also differentially expressed between the groups. Finally, *RAC1* (Fig. 6e, Supplementary Data 7E) is a member of the RAS pathway which has been shown to be associated with MYC pathway in CRPC samples[27]. Cox proportional hazards model analysis[38] demonstrated that 10 of the transcriptomic markers (i.e., *EIF6, ACAT1, TK1, PPP1R14B, TMEM54, UBE2S, DYNLL1, TUBA1C, RAC1, and WNT5A*) had significant association with response to Enzalutamide (Wald $p$ value < 0.05, Supplementary Data 7E, see Methods), yet many of them were borderline significant and none of them outperformed NME2 and MYC (Supplementary Data 7E). None of the genomic markers of Enzalutamide-specific response ($n = 2$), (described by refs. 18,109) were significantly enriched in the high-MYC and high-NME2 group (Fig. 6f, Supplementary Data 7F) or independently associated with response to Enzalutamide. Taken together, these findings indicate that the majority of the markers of PCa aggressiveness and therapeutic response that are enriched in the high-MYC and high-NME2 group are associated with MYC-related mechanisms and none of them outperform the ability of MYC and NME2 to predict risk of Enzalutamide resistance.

## Comparison to gene-centric computational methods

To evaluate if TR-2-PATH mechanism-centric predictions (i.e., activity levels of NME2 TR and MYC pathway) outperform predictive ability of commonly used gene-centric methods, we compared TR-2-PATH to differential expression analyses, Random (survival) Forest (RF)[110], and Support Vector Machine (SVM)[111] methods all utilized on the Enzalutamide-specific Abida et al. cohort. For differential gene expression analysis, we considered genes that were differentially expressed between the three phenotypes (Intact, EnzaSens, and EnzaRes) in the mining step I and considered genes at (i) Welch t-test $p$ value < 0.05; (ii) top 470 differentially expressed genes (comparable to the total number of target genes and pathway genes used for activity estimation) and *not excluding* target/pathway genes from NME2 TR and MYC pathway; (iii) top 470 differentially expressed genes, *excluding* target/pathway genes from NME2 TR and MYC pathway. For RF and SVM analysis, we utilized 470 genes from (iii) to avoid overfitting and then selected top 10 most significant genes/features from the outputs. Final gene list from each of these analyses was subjected to Kaplan-Meier survival analysis[37] and Cox proportional hazards model analysis[38] (crude and adjusted for age and Gleason), which did not show a significant association with Enzalutamide-associated disease progression using log-rank test, Wald test, or crude/adjusted hazards models (Supplementary Fig. 9a–d, see Methods). Such analysis demonstrates that mechanisms identified by TR-2-PATH (NME2 TR and MYC pathway) have significant advantage in predicting the risk of Enzalutamide resistance, compared to commonly used gene-centric methods.

## Targeting MYC and NME2 is beneficial in Enzalutamide-resistant conditions

Given that NME2 and MYC are upregulated in both patients that are at risk of Enzalutamide resistance and patients that fail Enzalutamide, we experimentally evaluated the benefits of therapeutic targeting of MYC and NME2 in similar experimental conditions. For this, we utilized LNCaP and C42B cell lines, as our experimental systems in Enzalutamide-naïve and Enzalutamide-resistant (EnzaRes) conditions (see Methods). To target MYC, we used MYCi975, a small molecule that directly inhibits MYC activity[30]. To understand the dose-dependent effect of MYCi975 in Enzalutamide-naïve and Enzalutamide-resistant conditions, we performed dose-response assays in both LNCaP and C42B cell lines (Fig. 7a, Supplementary Fig. 10a, see Methods) with varying doses of both drugs. This analysis demonstrated a striking reduction of the cell viability when treated with MYCi975 both in Enzalutamide-naïve and Enzalutamide-resistant conditions in a dose-dependent manner (Fig. 7a, Supplementary Fig. 10a).

Next, we utilized identified sub-$IC_{50}$ concentrations of Enzalutamide (10 μM) alone, MYCi975 (2 μM) alone, or Enzalutamide and MYCi975 in combination, to perform colony formation assay using Enzalutamide-resistant LNCaP and C42B cells (see Methods). Inhibition of MYC using MYCi975 reduced the colony formation of LNCaP-EnzaRes cells (one-tailed Welch t-test $p$ value = 0.013, Supplementary Fig. 10b) and C42B-EnzaRes cells (one-tailed Welch t-test $p$ value = 3.98E-6, Fig. 7b), compared to Intact (DMSO). Interestingly, the colony formation ability was significantly reduced when MYCi975 and Enzalutamide were administered in combination on LNCaP-EnzaRes cells (one-tailed Welch t-test $p$ value = 6.39E-5, Supplementary Fig. 10b) and C42B-EnzaRes (one-tailed Welch t-test $p$ value = 4.7E-8 Fig. 7b) compared to Intact (DMSO).

To evaluate the impact of Enzalutamide alone, MYCi975 alone, or in combination on the migratory capacity of Enzalutamide-resistant cells we performed Boyden chamber-based in vitro migration assay (see Methods) using C42B-EnzaRes cells (LNCaP cells do not migrate) and observed a significant reduction in cell migration when treated with MYCi975 (one-tailed Welch t-test $p$ value = 0.02, Fig. 7c, see Methods) and even greater reduction when treated with a combination of MYCi975 and Enzalutamide (one-tailed Welch t-test $p$ value = 0.003, Fig. 7c, see Methods). Positive control for MYC protein level after treatment with MYCi975 in C42B-EnzaRes cells was performed after 24 h of treatment (Supplementary Fig. 11).

Subsequently, to evaluate the effect of NME2 silencing on MYC expression, we first evaluated the expression of NME2 in LNCaP/C42B-EnzaRes cells, compared to wild-type LNCaP/C42B cells, which showed elevated levels of NME2 in both LNCaP-EnzaRes cells (Supplementary Fig. 10c, one-tailed Welch t-test $p$ value = 0.00188, see Methods) and C42B-EnzaRes cells (Fig. 7d, one-tailed Welch t-test $p$ value = 0.0005, see Methods). NME2 knockdown in C42B-EnzaRes cells using two different siRNAs (Fig. 7e, see Methods), demonstrated a significant reduction in expression of NME2 (Fig. 7e, left) and MYC (Fig. 7e, right), supported by the previously identified NME2 upstream regulation of MYC[112–114].

Finally, to evaluate if silencing of NME2 could enhance the negative effect of MYC inhibition on tumor cell metastatic potential, we performed Boyden chamber-based in vitro migration assay using C42B-EnzaRes cells (see Methods), which demonstrated that cell migration was further reduced when NME2 knockdown was added to MYCi975 administration (Fig. 7f).

To evaluate whether NME2 tumor expression has a direct role in promoting resistance to Enzalutamide in vivo, we sought to test the hypothesis that loss of NME2 in Enzalutamide-resistant C42B cells is sufficient to restore Enzalutamide sensitivity. First, we tested this in vitro, by generating CRISPR KO of NME2 in the C42B-EnzaRes cells (Supplementary Fig. 12) and subjecting them to Enzalutamide treatment for 72 h. The cells with impaired NME2 expression displayed increased sensitivity to Enzalutamide, compared to cells transfected with non-targeting sgRNA control (Fig. 7g).

To test this hypothesis in vivo, we engineered C42B-EnzaRes cells with a doxycycline-inducible shNME2 knockdown construct (C42B-EnzaRes Dox/shNME2). We validated NME2 knockdown and the role of NME2 in regulating MYC by measuring MYC protein levels after NME2 knockdown by Western blot (Fig. 7h). Mice were implanted with the C42B-EnzaRes Dox/shNME2 cells. When tumors were established, mice were subjected to Enzalutamide treatment in the presence or absence of Doxycycline in the drinking water or the appropriate control treatments. Enzalutamide alone had a minimal effect under the tested dosing conditions (Fig. 7i, Supplementary Fig. 13), while NME2 knockdown (with Doxycycline in drinking water) re-sensitized tumors to Enzalutamide, with an average tumor growth inhibition of 62%. Taken together, these results using relevant pre-clinical models of Enzalutamide resistance indicate that therapeutic targeting of the NME2/MYC axis is beneficial in Enzalutamide-resistant conditions and MYC inhibition could be combined with concurrent Enzalutamide administration for improved efficacy.

## Discussion

To investigate the mechanisms of Enzalutamide resistance in CRPC patients, we have reconstructed a CRPC-specific mechanism-centric regulatory network that encodes relationships between molecular pathways and their upstream transcriptional regulatory programs (i.e., TR-2-PATH). Such network has been reconstructed in a genome-wide manner for CRPC patients and was mined using signatures of Enzalutamide sensitivity and resistance. Our TR-2-PATH approach has several advantages that distinguish it from previously utilized methods. First, its focus on mechanism-centrality, which connects molecular pathways to their upstream regulatory programs, opens a door for the discovery of more effective biomarkers and optimized therapeutic interventions. Second, through a network mining/interrogation step our method overcomes several important drawbacks present in commonly utilized statistical analyses, including the multi-collinearity of the TRs and their effective prioritization for enhanced therapeutic targeting. Finally, our approach constitutes a valuable community resource to investigate CRPC phenotypes and could potentially be applicable to other cancer types.

We have interrogated the mechanism-centric CRPC-specific regulatory network using signatures of Enzalutamide sensitivity and resistance and identified MYC pathway and its upstream transcriptional regulator NME2 as markers of increased risk of resistance to Enzalutamide - a discovery that promises to prioritize patients for Enzalutamide intervention. Further, we have demonstrated that therapeutic targeting of MYC and NME2 could potentially provide an effective strategy for high-MYC and high-NME2 patients at risk of developing resistance to Enzalutamide and/or as a salvage therapy for patients that fail Enzalutamide.

The MYC oncogene[30,115] is a prominent early response gene and is known as a "super-transcription factor", potentially regulating transcription of at least 15% of the entire genome[116] and includes a basic-region/ helix-loop-helix/ leucine-zipper (BR/HLH/LZ) motif at the C terminus and three conserved elements known as MYC boxes (MB) 1–3 at the N terminus[117]. To work as a transcription regulator, MYC interacts with another helix-loop-helix/ leucine-zipper protein, MAX, through the C-terminal BR/HLH/LZ motif to form a complex that binds to the conserved sequences (CACGTG) in the transcriptional regulatory regions of the target genes[118–120], known to be involved in proliferation[121], differentiation[122], cell cycle[121,122], metabolism[123], apoptosis[121], angiogenesis[124], and therapeutic resistance[23,27] across different cancers.

Interestingly, a recent study by ref. 20 observed that MYC targets are open for binding in Enzalutamide-resistant conditions[20], which prompted us to evaluate MYC TR activity in the Enzalutamide-

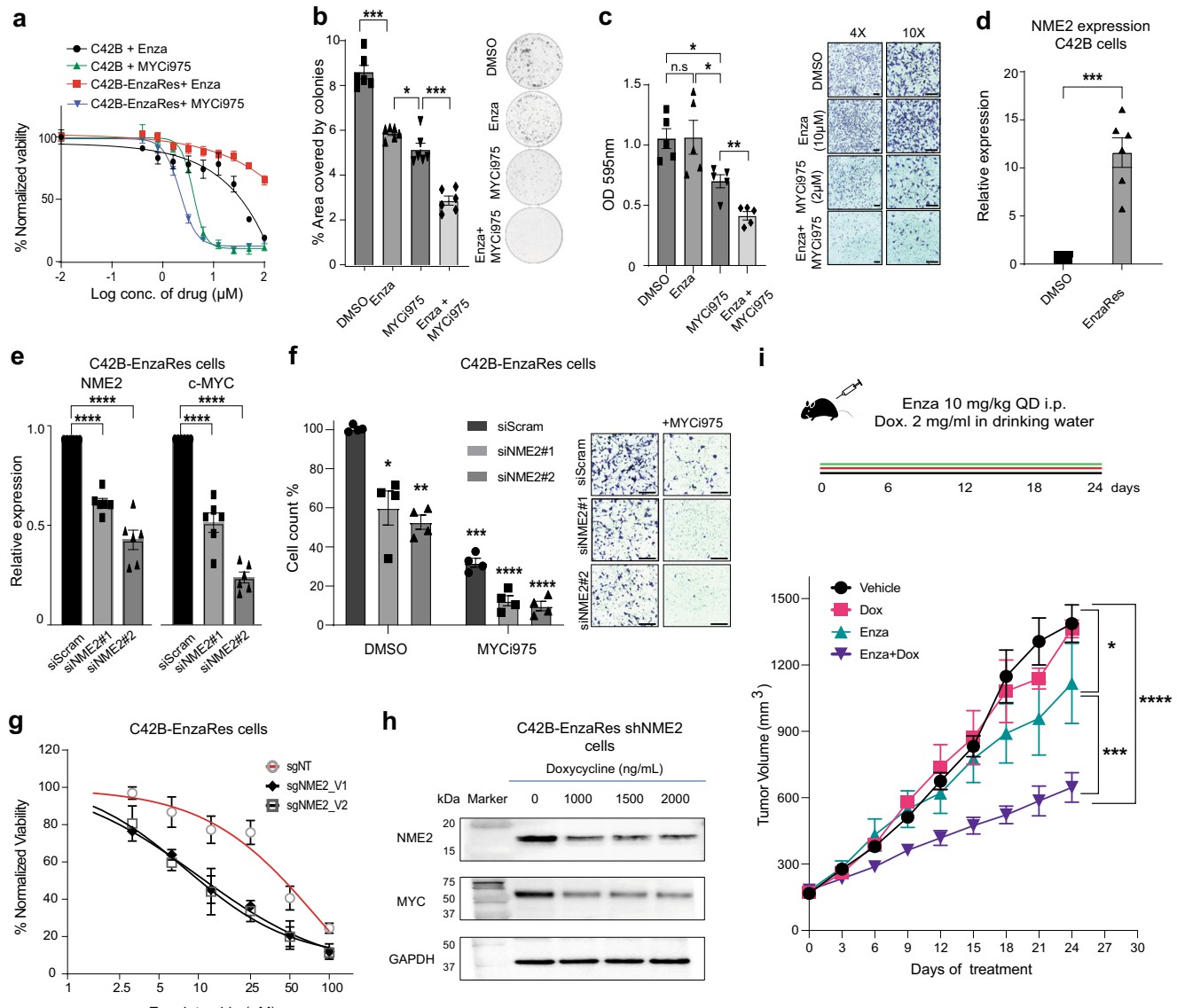

**Fig. 7 | MYC targeting is beneficial in Enzalutamide-resistant conditions. a** Drug sensitivity curves of Enzalutamide-naïve, or Enzalutamide-resistant (EnzaRes) C42B cells treated with MYCi975. Data are presented as mean values ± SEM from n = 3 biologically independent experiments. **b** Colony formation assay using Enzalutamide-resistant (EnzaRes) C42B cells in Intact (i.e., treated with DMSO), treated with Enzalutamide (10 μM), MYCi975 (2 μM), or a combination of Enzalutamide+MYCi975 (10 μM + 2 μM). Cells were grown in the presence of respective drugs. Representative images are shown, and data are presented as mean values ± SEM from *n* = 6 biologically independent experiments. *P* value was estimated using a one-tailed Welch t-test. **p* value < 0.05, ***p* value ≤ 0.001. **c** Boyden **c**hamber-based in vitro migration assay using Enzalutamide-resistant (EnzaRes) C42B cells in Intact (i.e., treated with DMSO), treated with Enzalutamide (10 μM), MYCi975 (2 μM), or a combination of Enzalutamide+MYCi975 (10 μM + 2 μM). Representative images are shown, data are presented as mean values ± SEM from n = 5 biologically independent experiments, indicating the quantification of Crystal Violet trapped by migrated cells. Scale bars, 100 μm. *P* value was estimated using a one-tailed Welch t-test. **p* value < 0.05, ***p* value ≤ 0.01. **d** Expression of NME2 in Intact and Enzalutamide-resistant (EnzaRes) C42B cells, using qRT-PCR, data are presented as mean values ± SEM from *n* = 6 biologically independent experiments. *P* value was estimated using the one-tailed Welch t-test. ****p* value ≤ 0.001. **e** Two different siRNAs targeting NME2 were used to downregulate NME2 (left panel) and its effect on MYC expression using qRT-PCR is shown (right panel), data are presented as

mean values ± SEM from *n* = 6 biologically independent experiments. *P* value was estimated using one-tailed Welch t-test. * *p* value < 0.05. **f** Boyden chamber-based in vitro migration assay using Enzalutamide-resistant (EnzaRes) C42B cells treated with DMSO or MYCi975 (2 μM) with or without knockdown of NME2. Representative images are shown, data are presented as mean values ± SEM from n = 4 biologically independent experiments, indicating the cell count quantification of Crystal Violet trapped by migrated cells, normalized to control (DMSO with siScramble). Scale bars, 100 μm. *P* value was estimated using 2-way ANOVA, **p* value < 0.05, ***p* value < 0.01, ****p* value < 0.001, *****p* value < 0.0001. **g** Comparison of anti-proliferative effects of Enzalutamide on C42B EnzaRes cells that have NME2 knocked down via CRISPR (sgNME2_V1 and sgNME2_V2) versus C42B EnzaRes which have received non-targeting sgRNA control (sgNT). Data are presented as mean values ± SEM from *n* = 3 independent biological replicates for each cell line. **h** Western blot of NME2, MYC and GAPDH protein levels in C42B shNME2 EnzaRes cells treated with Doxycycline for 96 h at the indicated concentrations. (n = 3 biologically independent experiments, representative blot shown). **i** Schematic representation and tumor volumes for mice bearing established C42B EnzaRes shNME2 xenografts treated with vehicle (*n* = 7 mice), Enzalutamide (10 mg/kg QD i.p) and/or Doxycycline (2 mg/mL in drinking water) (*n* = 6 mice for Dox, Enza, Enza + Dox arms) for 24 days. Statistical significance was performed via two-way ANOVA. **p* value < 0.05, ****p* value ≤ 0.001, *****p* value ≤ 0.0001. Source data for all the panels in this figure are provided as a Source Data file.

associated Abida et al.[33] cohort. When MYC TR activity was utilized as a predictive factor, we observed its modest ability to identify patients at risk of resistance to Enzalutamide (log-rank $p$ value = 0.01), yet at a smaller scale compared to the MYC pathway and its NME2 TR partnership (log-rank $p$ value = 0.0035), indicating a potential cross-talk between MYC transcriptional regulatory machinery and the MYC pathway.

Several groups have been working on direct and indirect strategies to target MYC-related mechanisms[30,125,126], with ref. 30 developing a small-molecule MYCi975 that directly inhibits MYC through disrupting MYC/MAX interaction (i.e., heterodimerization of MYC with MAX) to suppress the expression of MYC target genes and reduce MYC protein stability by enhancing phosphorylation of threonine 58. Administration of MYCi975 significantly reduces MYC-dependent cancer-cell proliferation, expression of MYC target genes, and tumor growth. Here, we demonstrate that administration of MYCi975 is beneficial in high-MYC Enzalutamide-resistant conditions and constitutes a potential therapeutic option for high-MYC CRPC patients who are at risk of Enzalutamide-resistance and/or who failed Enzalutamide. Our results agree with recent observations by Holmes et al.[127] that demonstrated MYCi975 regulates AR, AR-V7 and FOXA1 and sensitizes AR-V7+, Enzalutamide-resistant 22Rv1 tumors to Enzalutamide. Furthermore, a positive correlation of MYC pathway activity levels with AR expression and activity, previously observed by refs. 26,128, has also been confirmed by our investigations. Since MYC is known to bind to the regulatory regions of AR[26], MYC targeting (alone or in combination with AR inhibitors) in AR-associated conditions is a promising therapeutic avenue for a high-MYC subset of CRPC patients.

Our approach identified NME2 as a transcriptional regulatory program upstream of MYC, involved in Enzalutamide resistance in CRPC. Previously, NME2 has been shown to bind to the nuclease-hypersensitive element (NHEIII) in the promoter region of MYC interacting with G-quadruplex structure[65] and increase MYC expression, nominating NME2 as a transcriptional regulator of MYC[65,129]. NME2 is a member of the NME family[65] that has been shown to be involved in tumorigenesis and treatment response across different cancer types. In particular, ref. 130 demonstrated that NME2 plays a vital role in maintaining stemness of gastric cancer stem cells by enhancing the expression of anti-apoptosis genes, ref. 131 demonstrated suppression of apoptosis via NME2-mediated miR-100 upregulation in the development of gastric cancer, and ref. 132 demonstrated that over-expression of NME2 is associated with acquired resistance to 5-fluorouracil in colorectal cancer, yet its role (alongside partnership with MYC) in Enzalutamide resistance in CRPC patients has not been explored yet.

Here, we propose that knockdown of NME2, alongside targeting of MYC, is beneficial in Enzalutamide-resistant conditions for patients with high-MYC and high-NME2 profiles. Recently, ref. 129 have shown that a small molecule stauprimide binds to NME2 and inhibits its nuclear localization, thus affecting MYC transcription in renal cancer cell lines RXF 393 and CAKI-1. While shown to be effective in renal-specific cells, a potent NME2 inhibitor in PCa/CRPC settings is yet to be discovered and would constitute an effective combination strategy for high-MYC and high-NME2 CRPC patients.

In conclusion, we propose that MYC-centric mechanisms could be effectively utilized as markers to identify CRPC patients at risk of resistance to Enzalutamide. Pre-screening patients prior to Enzalutamide administration could potentially enhance personalized therapeutic planning, preclude harmful side effects, and extend patient survival. Further, we nominate MYC and NME2 as potential therapeutic targets for CRPC patients with high-MYC and high-NME2 profiles who are at risk of resistance to Enzalutamide and/or as a salvage therapy for patients that have failed Enzalutamide treatment.

## Methods

### Computational methods

**Datasets utilized in this work.** Datasets utilized for network construction, mining, validation, and negative control analysis are summarized in Supplementary Data 1 (supplied separately).

(i) **Dataset to associate activity levels of MYC pathway with response to Enzalutamide:** To determine if increased activity levels of MYC pathway (i.e., HALLMARK_MYC_TARGETS_V2 pathway from Hallmark collection in MSigDB 3.0) were associated with Enzalutamide resistance, we utilized Enzalutamide-associated CRPC metastatic samples from ref. 33 cohort (fresh-frozen needle biopsies), profiled on Illumina HiSeq 2500 and downloaded from https://github.com/cBioPortal/datahub/tree/master/public/prad_su2c_2019. We specifically selected samples that at the time of biopsy (sample collection) were ARSI-naïve (i.e., not subjected to any ARSI treatment), treated with Enzalutamide after sample collection, and then followed up for Enzalutamide-associated disease progression ($n = 22$, one sample per patient, as described in ref. 33). In this sub-group, the mean age at diagnosis was 59 years with a standard deviation of 6.85, the mean age at biopsy was 67.6 years with a standard deviation of 8.3, and the mean prostate-specific antigen (i.e., PSA) was 189.4 ng/ml with a standard deviation of 526.18. Metastatic composition of this sub-group included lymph node ($n = 13$), bone ($n = 6$), lung ($n = 1$), other soft tissue ($n = 1$), and liver ($n = 1$) samples. We utilized Enzalutamide-associated disease progression, defined as the time on Enzalutamide treatment without being subjected to another agent such as taxane, as the clinical end-point (as defined and suggested in ref. 33).

(ii) **Dataset to associate activity levels of MYC pathway with response to Abiraterone:** To determine if elevated activity levels of MYC pathway (i.e., MYC pathway was utilized as a geneset from HALLMARK_MYC_TARGETS_V2 pathway in Hallmark collection in MSigDB 3, $n = 58$, Supplementary Data 2) were specifically associated with Enzalutamide (and not Abiraterone) resistance, we utilized Abiraterone-associated metastatic CRPC sample from ref. 33 cohort. We specifically selected samples that at the time of biopsy (sample collection) were ARSI-naïve (as above), treated with Abiraterone after sample collection, and then followed up for Abiraterone-associated disease progression ($n = 33$, one sample per patient) for negative control analysis. The mean age at diagnosis for this patient sub-group was 61.38 years with a standard deviation of 5.94, the mean age at biopsy was 66.73 years with a standard deviation of 7.02, and the mean PSA was 51.4 ng/ml with a standard deviation of 91.05. Metastatic composition of this sub-group included lymph node ($n = 18$), bone ($n = 11$), liver ($n = 2$), and other soft tissue ($n = 2$) samples. We utilized Abiraterone-associated disease progression, defined as the time on Abiraterone treatment, without being subjected to other agents such as taxane, as the clinical end-point (as defined and suggested in ref. 33).

(iii) **Dataset for network reconstruction:** To construct a mechanism-centric network, we utilized the Stand Up to Cancer (SU2C) East Coast cohort[32,33], profiled on Illumina HiSeq 2500 and downloaded from dbGaP phs000915.v2.p2. This cohort included metastatic CRPC samples, obtained as fresh-frozen needle biopsies. We examined 280 samples available at dbGaP, and to avoid any overlap with treatment-associated analysis in ref. 33 (which we have utilized in part for validation and in part for a negative control), we removed all SU2C East Coast cohort samples that were present in ref. 33 ($n = 29$). Subsequently, we also removed samples that were duplicated (i.e., when the same sample was sequenced by different facilities) and selected one sample per patient to avoid signal duplication for our final network-building. Our final cohort comprised 153 patients with a

mean age at diagnosis of 59.2 years with a standard deviation of 8.38, a mean age at biopsy of 66.1 years with a standard deviation of 8.07, and a mean PSA of 234.5 ng/ml with a standard deviation of 1574.4. Metastatic composition of this cohort included adrenal ($n = 1$), bone ($n = 39$), liver ($n = 26$), lymph node ($n = 57$), other soft tissue ($n = 19$), prostate ($n = 4$), lung ($n = 2$) and unknown origin ($n = 5$) samples. At the time of biopsy (sample collection), patients either were exposed to ARSI ($n = 67$), were ARSI-naïve ($n = 75$), were on treatment ($n = 4$), or their treatment was unknown ($n = 7$).

(iv) **Datasets for network mining:** For network mining (i.e., query/interrogation), we utilized LNCaP cell line samples from ref. 53 ($n = 12$), that were profiled with the HumanHT-12 v4 Expression BeadChip and downloaded from GEO GSE78201. These dataset included three phenotypes: (i) LNCaP cells subjected to DMSO (referred to as Intact to indicate that they were not subjected to Enzalutamide treatment) ($n = 4$); (ii) LNCaP cells subjected to Enzalutamide for 48 h and sensitive to it (referred to as Enzalutamide-sensitive, EnzaSens) ($n = 4$); and (iii) LNCaP cells subjected to Enzalutamide for 6 months and having developed resistance to it (referred as Enzalutamide-resistant, EnzaRes) ($n = 4$).

(v) **Datasets for clinical validation:** For validation purposes, we utilized (i) ref. 19 (ii) Enzalutamide-associated ref. 33, and (iii) SU2C West Coast[70,71] datasets. First, to confirm that upregulation of the NME2 transcriptional regulatory program and MYC pathway characterize Enzalutamide-naïve samples (i.e., before patients were exposed to Enzalutamide) from patients that were later exposed to Enzalutamide and eventually failed it, we selected two sequential single-cell samples from the same CRPC patient (01115655) from ref. 19 cohort. These samples were profiled on Illumina NextSeq 500 and downloaded from https://singlecell.broadinstitute.org/single_cell/study/SCP1244/transcriptional-mediators-of-treatment-resistance-in-lethal-prostate-cancer. In particular, the first sample was collected before the patient was subjected to Enzalutamide and second sample was collected after the same patient received Enzalutamide and developed resistance to it. Both samples were collected from the lymph-node metastatic site.

Finding from ref. 19 were confirmed in Enzalutamide-associated Abida et al.[33] cohort. Briefly, we selected a subset of patients that were ARSI-naïve at biopsy, treated with Enzalutamide after sample collection, and subsequently monitored for Enzalutamide-associated disease progression ($n = 22$, as described above).

Further, we validated the predictive ability of NME2 TR and MYC pathway in SU2C West Coast cohort[70,71], which comprises samples from CRPC patients (obtained from fresh frozen image-guided core needle biopsies), profiled on Illumina HiSeq 2500 or NextSeq 500, accessed through dbGap phs001648.v2.p1 and downloaded from GDC (https://portal.gdc.cancer.gov/projects/WCDT-MCRPC). The samples in this cohort were subjected to Enzalutamide and/or Abiraterone either before biopsy (sample collection) or after biopsy (sample collection). Subsequently, all patients were monitored for disease progression ($n = 83$, one sample per patient). The mean age for the patients in this cohort was 70.59 years with standard deviation of 8.14. The patients in this cohort were from various races, including, white ($n = 70$), Asian ($n = 2$), African American ($n = 5$) and unknown ($n = 6$). Metastatic composition of this cohort included bone ($n = 36$), liver ($n = 7$), lymph node ($n = 31$), and unknown ($n = 9$). Further, samples were obtained from patients who were either in M1b stage (i.e., when prostate cancer has spread to bone, $n = 36$) or M1c stage (i.e., when prostate cancer has spread to other parts of the body, $n = 47$). We utilized treatment-associated disease progression (defined as an increase in PSA level (minimum 2 ng/mL) that has risen at least twice in an interval of least one week or soft tissue progression (nodal and visceral) based on RECIST v1.1) as the clinical end-point (as defined and suggested in refs. 70,71).

(vi) **Additional datasets for validation of association between NME2 TR and MYC pathway activities:** To further validate the association between NME2 TR and MYC pathway activities, we utilized (i) Alumkal et al.[12]; (ii) Beltran et al.[68] and (iii) Kumar et al.[69] CRPC patient cohorts.

Alumkal et al.[12] cohort includes CRPC samples isolated by laser capture microdissection from frozen biopsies ($n = 24$), profiled on Illumina HiSeq 2500 or NextSeq 500, and downloaded from Supplementary material of the associated paper[12].

Beltran et al.[68] cohort includes CRPC samples ($n = 34$, neuroendocrine samples excluded) profiled on HiSeq 2500 and downloaded from dbGaP (phs000909.v1.p1). These samples were obtained at biopsy from different metastatic sites including adrenal ($n = 1$), bone (n = 11), liver ($n = 3$), lung ($n = 2$), lymph node ($n = 9$), prostate ($n = 7$), and skull base ($n = 1$). ref. 69 cohort includes CRPC samples obtained at rapid autopsy, profiled on Agilent 44 K whole human genome expression oligonucleotide microarray and downloaded from GEO (GSE77930). The samples were obtained from different metastatic sites which included liver ($n = 21$), lymph node ($n = 69$), lung ($n = 22$), bone ($n = 20$), retroperitoneal ($n = 4$), kidney ($n = 1$), appendix ($n = 1$), peritoneum ($n = 2$), adrenal ($n = 4$), bladder ($n = 5$), bladder neck ($n = 2$), pelvic mass ($n = 1$), peritoneal ($n = 1$), renal ($n = 1$), scrotum ($n = 1$), skin ($n = 1$), spleen ($n = 1$).

(vii) **Datasets for negative control analysis:** To evaluate if the predictive ability of NME2 TR and MYC pathway are indeed Enzalutamide specific, we utilized (i) Abiraterone-associated ref. 33 cohort (as described above); and (ii) PROMOTE[34] cohort, as negative controls. As described above, Abiraterone-associated ref. 33 cohort included ARSI-naïve CRPC samples obtained at biopsy, treated with Abiraterone after sample collection, and subsequently monitored for Abiraterone-associated disease progression ($n = 33$, as described above).

PROMOTE[34] cohort included samples from patients with CRPC profiled on Illumina HiSeq 2500 and downloaded from dbGaP phs001141.v1.p1. These samples were obtained at biopsy from different metastatic sites ($n = 77$, one sample per patient), including bone ($n = 56$), soft tissue ($n = 2$), liver ($n = 2$), prostate bed ($n = 2$), lymph-node ($n = 14$), and lung ($n = 1$) and were ARSI-naïve at the time of sample collection. After sample collection, the patients were subjected to Abiraterone for 12 weeks, and were assessed for Abiraterone-associated disease progression right after that, which was defined based on the score that combined serum PSA level, bone and CT imaging, and symptom assessment at week 12. Patients that developed disease progression at week 12 were classified as non-responders ($n = 32$) and those that did not develop disease progression at week 12 were classified as responders ($n = 45$).

**Data download, processing, and normalization.** Abida et al.[33] RNA-seq samples profiled on Illumina HiSeq 2500, were downloaded from https://github.com/cBioPortal/datahub/tree/master/public/prad_su2c_2019 as Fragments Per Kilobase of transcript per Million mapped reads (FPKM). The clinical and treatment data were downloaded from the supplementary material of the corresponding paper[33] and from cBioPortal (https://www.cbioportal.org/).

SU2C East Coast[32] cohort RNA-seq samples profiled on Illumina HiSeq 2500, were requested and downloaded from dbGaP phs000915.v2.p2 as SRA files using the *prefetch* command and were converted to FASTQ files utilizing the *fastq-dump* command from *sra*

*toolkit* (version 10.8.2)[133]. Following this, the FASTQ files were aligned to a reference genome hg19 using STAR aligner[134] with the *quantMode* option, which generated raw count files. The raw counts were normalized using R *DESeq*[135] package for further statistical analysis. The clinical data were obtained from the supplementary material of the corresponding paper[33] and from cBioPortal (https://www.cbioportal.org/).

Kregel et al.[53] LNCaP cell line samples were profiled on HumanHT-12 v4 Expression BeadChip Kit and their quantile-normalized[136] gene expression data were downloaded from GEO GSE78201. The phenotype information was obtained from GEO GSE78201.

He et al.[19] single-cell RNA-seq samples profiled on Illumina Next-Seq 500, were downloaded from https://singlecell.broadinstitute.org/single_cell/study/SCP1244/transcriptional-mediators-of-treatment-resistance-in-lethal-prostate-cancer, as single-cell Transcripts Per Million (TPM) data matrix. The clinical data were obtained from the main body and supplementary material of the corresponding paper[19].

SU2C West Coast[70] cohort RNA-seq samples profiled on either Illumina HiSeq 2500 or NextSeq 500 were downloaded from GDC (https://portal.gdc.cancer.gov/projects/WCDT-MCRPC) as BAM files. These BAM files were then converted to FASTQ files utilizing *bam2-fastq* from *bedtools*[137]. Subsequently, the FASTQ files were aligned to a reference genome hg19 using STAR aligner[134] with the *quantMode* option, which generated raw count files. The raw counts were normalized using R *DESeq*[135] for further statistical analysis. The clinical and treatment data were obtained from GDC (https://portal.gdc.cancer.gov/projects/WCDT-MCRPC).

Alumkal et al. CRPC samples profiled on either Illumina HiSeq 2500 or NextSeq 500 were downloaded from the Supplementary file of the corresponding paper[12] as Transcripts Per Million (TPM) data matrix. The clinical file was obtained from the main body and Supplementary Material of the corresponding paper[12].

Beltran et al. cohort RNA-seq samples profiled on Illumina HiSeq 2500 were requested and downloaded from dbGaP phs000909.v1.p1 as SRA files using the *prefetch* command and were converted to FASTQ files utilizing the *fastq-dump* command from *sra toolkit* (version 10.8.2)[133]. Following this, the FASTQ files were aligned to a reference genome hg19 using STAR aligner[134] with the *quantMode* option, which generated raw count files. The raw counts were normalized using R *DESeq*[135] package for further statistical analysis. The clinical data were obtained from dbGaP phs000909.v1.p1.

Kumar et al. cohort CRPC samples profiled on Agilent 44 K whole human genome expression oligonucleotide microarray and their gene expression data were obtained from GEO (GSE77930). The clinical data was obtained from GEO (GSE77930).

PROMOTE[34] RNA-seq samples profiled on Illumina HiSeq 2500, were requested and downloaded from dbGaP phs001141.v1.p1 as SRA files using the *prefetch* command and then converted to FASTQ files using the *fastq-dump* command from *sra toolkit* (version 10.8.2)[133]. Subsequently, the FASTQ files were aligned to the reference genome hg19 using STAR aligner[134] with the *quantMode* option to generate raw count files. The raw count files were normalized using R *DESeq*[135] package. The clinical data were obtained from dbGaP phs001141.v1.p1.

**Estimating activity levels of molecular pathways.** A list of molecular pathways and their corresponding genes were obtained from Molecular Signatures Database (MSigDB)[138], available from Broad institute, and included C2 pathway collection (KEGG[41], BioCarta[42], and Reactome[43]) and Hallmark[44] gene sets. To estimate activity levels of each molecular pathway, we utilized signature-based or single-patient (i.e., single-sample) based Gene Set Enrichment Analysis (GSEA)[36], similarly to refs. 139,140. For the signature-based GSEA analysis, a signature of interest (e.g., defined as a list of genes ranked by their differential expression using two-tailed Welch t-test between any two phenotypes of interest, such as Enzalutamide-resistant and Enzalutamide-sensitive phenotypes) is used as a

reference signature and genes from a specific pathway are used as a query gene set. For single-patient (i.e., single-sample) GSEA analysis, gene expression profiles were scaled/standardized (i.e., z-scored) on gene-level so that mean of values for each gene was 0 and the standard deviation was 1, allowing for comparison of gene ranks across different samples. A single-sample signature was defined as a list of genes ranked by their z-scores and utilized as a reference signature in single-sample GSEA analysis (pathway genes were utilized for query, in the same manner as above). For signature-based and single-sample GSEA analysis, Normalized Enrichment Score (NES) and *p* values were estimated using 1000 gene permutations. NESs from this analysis were utilized as pathway activity values, where positive NES corresponds to an enrichment of pathway genes in the over-expressed part of the signature and negative NES corresponds to an enrichment of pathways genes in the under-expressed part of the signature.

**Estimating activity levels of transcriptional regulatory programs.** To estimate the activity levels of transcriptional regulators we utilized MARINa[141] (for a signature-based analysis) and VIPER[45] (for a single-sample-based analysis). Signatures were defined in the same manner as for the pathway enrichment analysis and were utilized as a reference for MARINa/VIPER. Instead of utilizing pathway data, MARINa and VIPER analyses require tissue-specific prostate cancer transcriptional regulatory network (interactome), as reconstructed previously in ref. 39 This interactome comprises of transcriptional regulators (TR, transcription factors and co-factors) and their potential transcriptional targets, connected by the transcriptional regulatory relationships. During MARINa/VIPER analysis, these transcriptional targets (for each transcriptional regulator separately) are utilized as a query gene set. We refer to the TR and the set of its corresponding transcriptional targets as a transcriptional regulatory program. Similar to GSEA, NESs/z-scores from MARINa and VIPER analysis were utilized to define activity levels of TRs. MARINa was implemented using *msviper* function and VIPER was implemented using *viper* function from R *VIPER* package in Bioconductor[45].

**TR-2-PATH: reconstruction of a mechanism-centric regulatory network.** To identify potential regulatory relationships between molecular pathways and their upstream transcriptional regulatory programs in CRPC patients, we have reconstructed a CRPC-specific mechanism-centric regulatory network using TR-2-PATH method. In this network, each node represents a mechanism: a molecular pathway or transcriptional regulatory program. SU2C East Coast cohort (as described above) was first scaled/standardized on the gene level and then subjected to single-sample pathway enrichment analysis (as described above) and single-sample transcriptional regulatory analysis (as described above). We then defined activity vectors for each molecular pathway (where each pathway vector corresponds to the NESs for this pathway across all patients in the SU2C East Coast cohort) and for each TR program (where each TR vector corresponds to the NESs/z-scores for this TR across all patients in SU2C East Coast cohort). Specifically, let us assume that we have *n* samples. If the activity level of pathway $i$ in sample $j$ is $NES_{i,j}$, then the activity vector for pathway $i$, $P_{i\,activity}$ is defined as.

$$P_{iactivity} = \begin{bmatrix} NES_{i,1} \\ NES_{i,2} \\ . \\ . \\ . \\ NES_{i,n} \end{bmatrix} \quad (1)$$

Similarly, if the activity level of a TR $t$ in a sample $j$ is $a_{t,j}$, then the activity vector for TR $t$, $TR_{t\ activity}$ is defined as

$$TR_{tactivity} = \begin{bmatrix} a_{t,1} \\ a_{t,2} \\ . \\ . \\ . \\ a_{t,n} \end{bmatrix} \quad (2)$$

To estimate potential regulatory relationships between transcriptional regulatory programs and molecular pathways, we first performed a pairwise comparison of each TR activity vector and each pathway activity vector using linear regression analysis, where a TR activity vector was used as a predictor variable (independent variable) and pathway activity vector was used as a response variable (dependent variable), as below. For each pathway $i$ and TR $t$:

$$P_{iactivity} = \alpha + \beta * TR_{tactivity} \quad (3)$$

The positive Beta ($\beta$) coefficient from the linear regression analysis (which corresponds to a positive slope for the fitted line between TR activity vector and pathway activity vector) indicated a positive relationship/association from the TR to the pathway and a negative Beta coefficient (negative slope) indicated a negative relationship/association from the TR to the pathway. Following the regression analysis for all TR-pathway pairs, we subjected it to multiple hypotheses FDR correction, which was performed for each pathway separately. If this relationship showed FDR < 0.05, it was added as an edge to the final network. Otherwise, it was discarded. Linear regression analysis was performed using the R *lm* function[142] and multiple hypotheses testing per pathway was performed using the R *p.adjust* function[142].

**Bootstrap analysis for the mechanism-centric regulatory network.** To evaluate if the edges in the mechanism-centric regulatory network could be "recovered" in the presence of noise (re-sampling), we performed bootstrap analysis. For this, SU2C East Coast cohort gene expression profiles ($n = 153$) were sampled with replacements 100 times. Each sampled/bootstrapped gene expression profile was then used to reconstruct a bootstrapped mechanism-centric regulatory network using the TR-2-PATH method (as above). We then utilized results from these 100 networks to assign weights to each edge, which was defined as the number of times this edge appeared (i.e., was recovered) across 100 bootstrapped networks (i.e., edge frequency). In particular, the edge weights were defined as the percent (%) of times an edge identified in the original network was also identified across the bootstrapped networks while maintaining the same direction of the relationship (positive/negative) between a particular TR program and a particular molecular pathway, across all 100 bootstrapped networks. These edge weights were then added to the original network (making it a weighted mechanism-centric network) and further utilized in the network query step.

The R functions *hist* and *density*[142] were utilized to depict weight distributions. To cluster the molecular pathways based on their edge weights, we utilized t-distributed stochastic neighbor embedding clustering[47] (t-SNE), a common dimensionality reduction technique that clustered pathways with similar edge weight patterns as nearby points and pathways with dissimilar edge weight patterns as distal points. t-SNE was implemented using the *Rtsne* function from R *Rtsne* package[143].

**Network mining I: identifying differentially altered sub-networks.** To identify parts of the mechanisms-centric network (i.e., sub-networks comprising of the molecular pathways and their upstream

TR programs) that significantly alter their activity across the response to Enzalutamide, we queried (i.e., mined) the mechanism-centric regulatory network using signatures of Enzalutamide-response. In particular, we specifically utilized gene expression profiles from ref. 53 (as described above), which consists of (i) Intact (DMSO subjected) LNCaP cells ($n = 4$), (ii) Enzalutamide-sensitive (EnzaSens) LNCaP cells ($n = 4$); and (iii) Enzalutamide-resistant (EnzaRes) LNCaP cells ($n = 4$). We hypothesized that if a particular sub-network is up-regulated (positive NES) in the intact state, then becomes down-regulated (negative NES) in the sensitive state, yet "recovers" and again become up-regulated (positive NES) in the resistant state (we call this "up-down-up" behavior), then such sub-network is important in Enzalutamide-resistance and could potentially constitute a functional marker and a therapeutic vulnerability. To identify such sub-networks and establish the significance of this change, we defined two gene expression query signatures (i) signature between intact and sensitive phenotype; and (ii) signature between sensitive and resistant phenotype. These signatures were defined utilizing two-tailed Welch t-test and implemented using the R *t.test* function[142].

To identify sub-networks with such "up-down-up" behavior, we evaluated their enrichment in the "intact to sensitive" signature (i.e., looking for "up-down" behavior, corresponding to the down-regulation as a result of response to Enzalutamide) and enrichment in the "sensitive to resistant" signature (i.e., looking for "down-up" behavior, corresponding to the subsequent up-regulation as a result of resistance to Enzalutamide). To achieve this, we first estimated pathway activity levels and TR activity levels in each signature and overlayed them with our mechanism-centric regulatory network relationships/structure to identify parts of the network that exercise "up-down-up" behavior, as described above. To estimate if such "up-down-up" changes were statistically significant, we performed pathway-on-pathway and TR-on-TR GSEA, where pathways from "intact to sensitive" signature were compared to pathways from "sensitive to resistant" signature (same for the TR programs). Parts of the network with significant negative enrichment in "intact to sensitive" signature and significant positive enrichment in "sensitive to resistant" signature (GSEA $p$ value < 0.001) were utilized for Network mining step II.

**Network mining II: prioritization of upstream regulatory programs Variance Inflation factor analysis.** Sub-networks identified in "Network mining I" include molecular pathways and their potential upstream TR programs. Such TR programs might exercise multi-collinearity in their effect on the pathway and could obstruct further statistical analysis (by making results not interpretable), yet deserve to remain in the analysis (as opposed to simply being eliminated). First, to check for multi-collinearity among TRs, we subjected the activity level of these TRs to Variance Inflation Factor analysis (VIF)[57] in the SU2C East Coast cohort. VIF runs a multivariable regression analysis, iteratively using each TR (activity vector) as a response variable and activity vectors from the rest of the TRs as predictor variables. The percentage of variation that the predictor variables could explain about the response variable is defined by the coefficient of determination, $R^2$, where higher $R^2$ values indicate a higher degree of multi-collinearity and VIF is defined as $1/(1 - R^2)$. Typically, the multi-collinearity is observed if VIF > 10. VIF analysis was implemented utilizing the *vif* function from the R *usdm* package[144].

**PLS regression analysis.** To address TR multi-collinearity, we developed a Partial Least Squares (PLS) -inspired method. To prioritize the effect of TR programs on a specific pathway $i$, our approach considers TR activity vectors $TR_{t\ activity}$, $t = 1...m$ (where $m$ is the number of TRs upstream of a specific pathway $i$) as predictor variables and utilizes a pathway $i$ activity vector $P_{i\ activity}$ as a response variable. TR activity vectors are then regressed (linear regression) on the pathway vector so

that their $\beta$ coefficients (slopes), indicating the effect of each TR on a pathway $i$, are denoted as weights $w_t$. Next, utilizing the TR activity vectors and weights associated with each TR, first latent variable $LV1$ is defined as:

$$LV1 = \sum_{t=1}^{m} TR_{t\,activity} * w_t \qquad (4)$$

Further, the contribution (also referred to as loadings) of each TR on the $LV1$ is determined through a multivariable regression analysis, where the activity vectors of all the transcriptional regulators $TR_{t\,activity}$ are utilized as independent variables and the $LV1$ utilized as a dependent variable. The $\beta$ coefficients associated with each TR in this multivariable analysis, indicating the contribution of each $TR_{t\,activity}$ to $LV1$, adjusted for the effect of all other TRs, are denoted as $loadings$. Loadings are most often utilized in social science analyses.

This latent variable $LV1$ is then "subtracted" from the TR activity vectors and the pathway $i$ activity vector, leaving the residuals to be utilized for defining the next latent variable. In particular, the first latent variable $LV1$ is utilized as an independent variable to be regressed on the activity vectors of each TR program $TR_{t\,activity}$ as well as activity of the molecular pathway $P_{i\,activity}$, so that the residuals from this analysis explain amount of information that has not been explained by $LV1$. The residuals are then utilized to define the second latent variable $LV2$ in the similar fashion. This process is repeated until latent variables can explain a significant amount of information about a pathway $i$. PLS was implemented utilizing the *plsreg1* function from the R *plsdepot* package[145].

**PLS-inspired circle of correlation analysis.** Identified latent variables do not express collinearity or multi-collinearity and are utilized as axes to build a "circle of correlation", which depicts the association of TR programs and a specific pathway $i$ (defined as arrows on the circle of correlation) to each latent variable. In particular, axes of the circle of correlation depict Pearson correlation $r$ values, defined between latent variables and TR/pathway activity vectors. Each TR and a pathway $i$ are indicated as arrows on the circle of correlation, with $x$ and $y$ coordinates that correspond to the values of Pearson correlation between their vectors and the latent variables.

To identify TRs that effect a specific pathway $i$ as a group, we developed a method that utilized unsupervised hierarchical clustering on the degree of closeness (angle) between TR and pathway arrows so that TRs in high proximity to one another (thus having similar effects on latent variables) are grouped as they express simultaneous effect on the pathway $i$. In particular, for each TR and pathway arrow we first calculated their angle of inclination i.e., $(\cos^{-1}\theta)$. To calculate the $\cos^{-1}\theta$ we utilized R *acos* function[142]. Following this, angle of inclination in radian was converted to a degree using the *rad2deg* function from the R *rCAT* package[146]. To determine the degree of closeness, we subtracted the angle of inclination of each TR arrow from angle of inclination of a pathway $i$ arrow. These degrees of closeness for TRs were then subjected to hierarchical clustering, which identified groups of TR programs with similar effects on the pathway $i$. For hierarchical clustering we utilized the R *hclust* function[142].

**Prioritizing TR groups.** The TR groups/clusters (which also include groups with one TR) are then "prioritized" based on their effect on a pathway $i$ using "effect scores", which are defined as a combination of (i) degree of closeness between a TR group/cluster and a pathway $i$ on the circle of correlation; (ii) association (i.e., Pearson correlation $r$) between a TR group/cluster and each evaluated latent variable; and (iii) edge weight between a TR group/cluster and a pathway $i$ from the TR-2-PATH mechanism-centric network reconstruction step. For clusters that contained more than one TR, average values for all TRs in that cluster were considered. Each of these categories assigned a "rank" for

each cluster and then ranks were combined (using geometric mean) to define the final effect score for each cluster. Geometric mean was implemented utilizing the *geometric.mean* function from the R *psych* package[147].

**Validation in independent cohorts and Enzalutamide specificity analysis.** For validation and negative control analysis, we utilized refs. 19, 33, SU2C West Coast[70,71], and PROMOTE[34] cohorts. Clinical characteristics and data normalization for these cohorts are described above and in Supplementary Data 1.

In He et al., (i.e., single-cell profiles), we reproduced data analysis performed in the original manuscript[19]. In particular, we applied UMAP[148] dimensionality reduction technique on single-cell Transcripts Per Million (TPM) data for each sample of the selected patient. We then utilized the AR activity and CK8 and CD45 expression on the UMAP projected data to identify adenocarcinoma cell clusters. Next, we estimated NME2 TR activity and MYC pathway activity on a single-cell level, in a manner similar to the single-sample analysis (as described above) and compared their activities between adenocarcinoma cells and the rest of the cells utilizing one-tailed Welch t-test, using the *t.test* function in R[142].

In Abida et al.[33], we subjected the cohort samples to a single-sample pathway and single-sample TR analysis to estimate activity levels of MYC pathway and NME2 TR program across all samples. For Enzalutamide-associated subset, we first performed Cook's distance[149] analysis to identify outliers that can influence the regression analysis results (no outliers identified) utilizing R *cooks.distance* function[142]. Following this, we performed association analyses between activity vectors of NME2 TR and MYC pathway using the R *cor.test* function[142]. Next to identify patients with high-NME2 TR and high-MYC pathway activities in Enzalutamide-associated subset, we performed hierarchical and kmeans clustering on MYC pathway and NME2 TR activity vectors. For Abiraterone-associated subset, we also performed the Cook's distance analysis (one outlier identified and removed) to identify outliers, followed by identification of patients with high-NME2 TR and high-MYC pathway activities. To identify patients with high-NME2 TR and high-MYC pathway activities, we applied the same thresholds that were estimated in the Enzalutamide-associated subset. Hierarchical clustering was implemented using the R *hclust* function[142] and kmeans clustering was performed using the R *kmeans* function[142] and identified two clusters of patients (i) patients with high-NME2 activity and high-MYC pathway activity and (ii) the rest of the patients (e.g., patients with low-NME2 and low-MYC pathway activity; patients with low-NME2 and high-MYC pathway activity; and patients with high-NME2 and low-MYC pathway activity). Further, to evaluate the difference in treatment response between the two identified groups, we utilized Kaplan-Meier survival analysis[37] and Cox proportional hazards model analysis[38], where treatment-associated disease progression (as described above) was utilized as the clinical end-points, as defined in ref. 33. For Kaplan-Meier survival analysis, we utilized the *Surv* and the *ggsurvplot* functions from R *survival*[150] and *survminer*[151] packages, respectively. The Cox proportional hazards model analysis was adjusted for age and Gleason score and utilized the *coxph* function from the R *survival* package[150].

In SU2C West Coast[70,71] cohort, similar to analysis on Abida et al., we subjected the cohort samples to a single-sample pathway and single-sample TR analysis to estimate activity levels of MYC pathway and NME2 TR program across all samples. As above, we first performed Cook's distance analysis[149] to identify outliers (three outliers identified and removed) using R *cooks.distance* function followed by performing association analyses between activity vectors of NME2 TR and MYC pathway using the R *cor.test* function. Next, to identify patients with high-NME2 TR and high-MYC pathway activities we utilized hierarchical and kmeans clustering on MYC pathway and NME2 TR activity vectors. Hierarchical clustering was implemented using R *hclust*

function and kmeans clustering was implemented using the R *kmeans* function and identified two clusters of patients (i) patients with high-NME2 activity and high-MYC pathway activity and (ii) the rest of the patients (e.g., patients with low-NME2 and low-MYC pathway activity; patients with low-NME2 and high-MYC pathway activity; and patients with high-NME2 and low-MYC pathway activity). Further, to evaluate the difference in treatment response between the two identified groups, we utilized Kaplan-Meier survival analyses[37] and Cox proportional hazards model analysis[38], where treatment-associated disease progression (as described earlier) was utilized as the clinical endpoints. For Kaplan-Meier survival analysis, we utilized the *Surv* and the *ggsurvplot* functions from the R *survival*[150] and *survminer*[151] packages respectively. The Cox proportional hazards model analysis was adjusted for race, Mstage, age and metastatic site and utilized the *coxph* function from the R *survival* package[150].

In PROMOTE[34] cohort, we first performed Cook's distance analysis[149] to identify outliers as above (two outliers identified and removed) using R *cooks.distance* function. Since PROMOTE cohort has binary outcomes (responders vs non-responders), to evaluate the ability of NME2 TR and MYC pathway activities to classify patients based on their binary response to Abiraterone treatment, we performed ROC analysis using a multiplicative logistic regression model[152], where the product of activity level of the NME2 TR program and activity level of the MYC pathway was utilized as predictor (independent) variable and responder/non-responder classification was utilized as response (dependent) variable. ROC curves were evaluated using area under the curve (AUROC), with AUROC = 0.5 being a random classifier. The logistic regression analysis was implemented using the R *glm* function[142] and ROC analysis was implemented using the *roc* function from the R *pROC* package[153].

**Comparison to markers of aggressiveness and therapeutic response.** To compare the ability of MYC and NME2 to predict Enzalutamide resistance to the predictive ability of known transcriptomic and genomic markers of aggressiveness and therapeutic response we utilized patients from Enzalutamide-associated Abida et al[33]. cohort (as described above). In particular, comparisons were done in two ways: (i) comparison between high-NME2 and high-MYC pathway patients and the rest of the patients ("others"), as described above using two-tailed Welch t-test (for transcriptomic markers) and Fisher exact test[154] (for genomic markers); and (ii) direct independent association with the Enzalutamide-associated disease progression using Cox proportional hazards model. For transcriptomic markers, we utilized their gene expression/normalized counts. For genomic markers, we utilized genomic alterations (obtained from cbioportal), including deep and shallow deletions, gains, and amplifications, as available in cbioportal.

Two-tailed Welch t-test was implemented using the R *t.test* function[142], Fisher exact test[154] was implemented using the R *fisher.test* function[142], and Cox proportional hazards model analysis was implemented using the *coxph* function from the R *survival* package[150].

**Comparative analysis of gene-centric computational methods.** To evaluate if TR-2-PATH mechanism-centric predictions (i.e., activity levels of NME2 TR and MYC pathway) outperform predictive ability of commonly used gene-centric methods, we compared TR-2-PATH to differential expression analyses, Random (survival) Forests (RF)[110], and Support Vector Machine (SVM)[111] methods all utilized on the Enzalutamide-associated Abida et al.[33] cohort. For differential gene expression analysis, we considered genes that were differentially expressed between the three phenotypes (Intact, EnzaSens, and EnzaRes) in the mining step I and considered genes at (i) Welch t-test *p* value < 0.05; (ii) top 470 differentially expressed genes (comparable to the total number of target genes and pathway genes used for activity estimation) and *not excluding* target/pathway genes from NME2 TR and MYC pathway; (iii) top 470 differentially expressed genes,

*excluding* target/pathway genes from NME2 TR and MYC pathway. For RF and SVM analysis, we utilized 470 genes from (iii) to avoid over-fitting and then selected top 10 most significant genes/features from the outputs. Final gene list from each of these analyses were utilized to cluster patients using hierarchical and kmeans clustering (as above), and then subjected these groups to Kaplan-Meier survival analysis[37] and Cox proportional hazards model analysis[38]. For Kaplan-Meier survival analysis, we utilized the *Surv* and the *ggsurvplot* functions from the R *survival*[150] and the *survminer*[151] packages, respectively. Additionally, for Cox proportional hazards model analysis, we utilized the *coxph* function from the R *survival* package[150]. For adjusted Cox proportional hazards model analysis, the model was adjusted for age and Gleason score. Random (survival) Forests were constructed utilizing *rfsrc* function from R *randomForestSRC* package[155]. The tuning parameters for Random (survival) Forests included (i) the maximum number of trees (i.e., "ntrees"), (ii) the number of variables assessed at each split (i.e., "mtry"), and (iii) maximum number of samples in the terminal (leaf) nodes (i.e., "nodesize"). The optimization of mtry and nodesize variables was performed utilizing *tune* function from R *randomForestSRC* package, which determined optimal value for mtry as 100 and nodesize as 5 and iterations of ntrees converged to a stable C-index around 3000, thus 3000 was selected as an optimal value for ntrees. For SVM, we utilized *fit* function from R *rminer* package[156] with default parameters.

**Statistical analysis and data visualization for computational methods.** Statistical analysis was performed using R studio version 4.0.4 for statistical computing. For differential gene expression we utilized one-tailed Welch t-test, with t-values and *p* values reported, corrected for multiple hypotheses testing. Linear regression analysis was utilized to evaluate relationships between TR programs and molecular pathways (i.e., network edges); the significance that the slope coefficient is non-zero was estimated using two tailed t-test, with multiple hypothesis testing. To ensure the edges identified were robust to sampling noise, 100 bootstrapped networks were generated, and each edge was assigned a weight, defined as the number of times an edge identified in the original network was also identified across the bootstrapped networks while maintaining the same direction of the relationship (positive/negative) between a particular TR program and a particular molecular pathway, across all 100 bootstrapped networks. To evaluate compositional similarity between MYC pathway genes and target genes of AR/NME2 transcriptional regulatory programs, we utilized Jaccard similarity index, defined as:

$$Jaccard\ similarity\ index = \frac{number\ of\ common\ targets\ between\ A\ and\ B}{total\ number\ of\ targets\ in\ A\ and\ B}$$

(5)

where *A* depicts MYC pathway genes and *B* depicts target genes of either NME2 TR or AR TR.

Kaplan-Meier survival analysis was utilized to estimate difference in treatment response between two patient groups, with log-rank test utilized for significance. Cox proportional hazards model was utilized to evaluate association with therapeutic response and its significance was reported as hazard ratio, confidence interval, hazard *p* value and Wald test. Further, to compare the ability of MYC and NME2 to predict Enzalutamide resistance to the predictive ability of known transcriptomic and genomic markers of aggressiveness and therapeutic response we utilized two-tailed Welch t-test (for continuous variables) and Fisher exact test (for categorical variables). We utilized the *geom_violin* and the *geom_boxplot* function from the *ggplot2*[157] in R for data visualization.

## Experimental methods

**Generation of Enzalutamide-resistant cell lines.** LNCaP (clone FDG) and C42B cells were purchased from ATCC and were grown in RPMI 1640 media (GIBCO # 11875093) supplemented with 10% Fetal Bovine Serum (FBS, Corning Cat#35-011-CV) and maintained at 37 °C and 5% $CO_2$. Enzalutamide powder was purchased from Sellekchem (cat #S1250) and re-suspended in DMSO. Cells were plated in 6-well plates and treated either with DMSO, or with Enzalutamide (20 µM), refreshed every 4 days for up to 3 months until the resistance emerged. RNA from cells was extracted on indicated days using the methods described below.

**Generation of inducible shNME2 cell line and NME2 CRISPR KO cell line.** We constructed C42B EnzaRes Dox/shNME2 cells by infecting C42B EnzaRes cells with pLKO-Tet-On-shNME2 virus followed by continuous puromycin selection. The lentiviral shRNA plasmid was constructed by inserting target shNME2 oligonucleotide sequence into Tet-pLKO-puro (Addgene, #21915) plasmid. For generating NME2 CRISPR KO cells, sgRNA oligonucleotide sequences were cloned into pLentiCRISPR-v2 plasmid (Genscript).

For both NME2 KD and KO cells, virus was generated as follows: plasmid DNA was extracted using a DNA extraction kit (Vazyme, DC112–01). The lentivirus was packaged by transfecting the plasmids with packaging vectors (psPAX and pMD2.G) and Lipofectamine 2000 (Invitrogen, #11668-019) in Opti-MEM media (Gibco) into HEK293T cells. Afterward, the virus supernatant was collected, filtered with a 0.45 µm strainer, concentrated with PEG6000 (Sigma, #81253), resolved in PBS and then aliquoted for subsequent transfection. Cells were infected with viruses and selected for 72 h with puromycin (2 µg/mL, Sigma-Aldrich, P8833). The following oligonucleotide sequences were used: TCATGGCAGTGATTCAGTAAA for the shNME2 sequence (flanked by start ACCGGT and end GAATTC sequences for a total of 33 bp) and CACCTTCATCGCCATCAAGC for sgNME2_V1 and GCACT-CACCATGGCCACAAC for sgNME2_V2. Cells transfected with the second sgRNA guide, sgNME2_V2, were used for further in vitro studies.

**RNA extraction, cDNA preparation, transcript knockdown, and qRT-PCR analysis.** RNA was isolated from cells by the Quick-RNA miniprep kit (Zymogen# R1054) and digested with DNase (provided in the kit). cDNA was synthesized from 1 µg RNA, using an All-in-One 5X RT-master mix (Abm # G592), per the manufacturer's protocol. qRT-PCR was carried out on the StepOne Real-Time PCR system (Applied Biosystems) using gene-specific primers designed with Primer-BLAST and synthesized by IDT Technologies. ON-TARGETplus SMARTpool (cat# L005102-00-0005) was obtained from Dharmacon and was used at 100 nmol/L. Cells were transfected in 6-well plates at a density of 100,000 cells per well using Lipofectamine RNAiMax (Invitrogen #13778075), according to the manufacturer's protocol. RNA was extracted and converted to cDNA as described above. qRT-PCR data were analyzed using the relative quantification method using 18sRNA as an internal reference, and plotted as average fold-change compared with DMSO or the non-targeting siRNA (i.e., Relative Quantity or RQ). Determination of transcript levels was carried out using Fast SYBR Green Master Mix (Invitrogen), using specific primer sets for c-MYC: c-MYC (F) 5'- CCTGGTGCTCCATGAGGAGAC-3'; c-MYC (R) 5'- CAGAC TCTGACCTTTTGCCAGG.

**Evaluating expression of AR.** To evaluate the expression of AR in Enzalutamide-naïve and Enzalutamide-resistant conditions, we utilized cells from LNCaP and C42B cell lines under Enzalutamide-naïve and Enzalutamide-resistant conditions (as described above) and determined the expression level of AR under both conditions using qRT-PCR assay (described above). The specific set of primers used for AR includes: AR (F) 5'- TCTTGTCGTCTTCGGAAATGTT-3'; AR (R) 5'-AAGCCTCTCCTTCCTCCTGTA-3'.

**Evaluating expression of NME2 in Enzalutamide-resistant vs Enzalutamide-nave cells.** To evaluate the expression of NME2 in Enzalutamide-naïve and Enzalutamide-resistant conditions, we utilized cells from LNCaP and C42B cell lines under Enzalutamide-naïve and Enzalutamide-resistant conditions (as described above) and determined the expression level of NME2 under both conditions using qRT-PCR assay (as described above). The specific set of primers used for NME2 is: NME2 (F) 5'- AGGATTCCGCCTTGTTGGTCTG-3'; NME2 (R) 5'-CGGCAAAGAATGGACGGTCCTT-3'.

**Knockdown of NME2.** Two different siRNA against NME2 (siNME2#1 AAUAAGAGGUGGACACAAC; siNME2#2 CUGAAGAACACCUGAAGCA), or non-targeting control (siScram) were obtained from Dharmacon and used at 100 nmol/L.

**Drug response curves.** Treatment-naïve or Enzalutamide-resistant C42B and LNCaP cells were plated in 96-well plates at a density of 5000 cells/well. A day later, cells were treated with indicated doses of drugs (6 technical replicates/dose) and were incubated for 96 h. After 96 h, cell viability was determined using CCK-8 reagent kit (Bimake # 34304) according to manufacturer's protocol. Graphs were plotted using GraphPad PRISM software and $IC_{50}$ values were determined. The experiment was repeated three times.

**Colony formation assay.** C42B-EnzaRes and LNCaP-EnzaRes cells were treated with siRNAs, or Enzalutamide or MYCi975, or a combination of both (Enzalutamide + MYCi975) as indicated for 24 h. Cells were then trypsinized (TrypLE Express, GIBCO # 12604013) and live cells were counted using Trypan Blue exclusion method. Five thousand live cells per condition were plated in 60 mm plates (in duplicates) and were grown in complete media with or without drugs as indicated. Cells were allowed to form colonies for two weeks, following which, the resulting colonies were fixed using 4% paraformaldehyde (Sigma Aldrich) for 30 min. Cells were washed with PBS and stained with 0.25% Crystal Violet for 10 min. Plates were washed with PBS to wash off unbound Crystal Violet. The percentage of area occupied by colonies was calculated by ImageJ software and graphs were plotted in Graph-Pad PRISM software. The experiment was repeated three times.

**Boyden chamber-based migration assay.** Boyden transwell chambers for migration were purchased from Corning Inc. (Cat# 353097). C42B-EnzaRes cells were transfected with siRNAs as described above, or treated with 2 µM MYCi975[30] and/or 10 µM Enzalutamide for 24 h. Cells were then trypsinized and re-suspended in serum-free media at a density of 100,000 cells/chamber, in the upper chamber in duplicates. The lower chamber was supplied media containing 10% FBS (in the case of NME2 knockdown), or supplemented with 10% FBS containing DMSO, or 2 µM MYCi975 (ref) and/or 10 µM Enzalutamide wherever indicated. Cells were allowed to migrate for 48 h, following which they were fixed using 4% paraformaldehyde for 30 min at room temperature. Wells were washed and membranes were stained with 0.25% crystal violet made in 25% methanol for 20 mins. Wells were washed thoroughly, and images were taken on an Echo-Revolve-R4 microscope. For quantification, cells stained with crystal violet were quantified using the image processing software suite ImageJ. Graphs were plotted using GraphPad Prism software. The experiment was repeated three times.

**Western Blot analysis.** Cells were lysed in RIPA (Sigma) lysis buffer containing 1x Halt™ Protease and Phosphatase Inhibitor Cocktail (ThermoFisher). Protein concentration was measured by Bio-Rad Bradford reagent. Protein samples were prepared by addition of 4x Laemmli Sample buffer (Bio-Rad) and 2-mercaptoethanol (Bio-Rad) and resolved on 4–12% SDS-PAGE (Sodium dodecyl sulfate–polyacrylamide) gels, which were subsequently transferred to

PVDF membranes (Bio-Rad) using Trans-Blot Turbo transfer buffer (Bio-Rad) and a Trans-Blot Turbo Transfer System (Bio-Rad). Membranes were blocked for 1 h at room temperature with 5% blotting-grade blocker non-fat dry milk (Bio-Rad), followed by overnight 4 °C incubation with the appropriate primary antibody and 1 h room temperature incubation with an anti-rabbit or anti-mouse IgG (H + L)-HRP conjugate (Bio-Rad) secondary antibody. Blots were imaged using Supersignal West Femto Maximum Sensitivity Substrate detection system (Thermo) and the ChemiDoc Imaging System (Bio-Rad). The following primary antibodies were used: c-Myc (Y69) (Abcam #ab32072, 1:1000 dilution), NME2 (4G7A8) (Abcam #ab204958, 1:1000 dilution), GAPDH (Cell Signaling Technology #3683, 1:5000 dilution), Actin (Cell Signaling Technology #5125 S, 1:5000 dilution).

**In vivo studies.** All animal experiments and procedures were performed in compliance with ethical standards and the approval of Northwestern University Animal Care and Use Committee (IACUC). All mice were obtained from Jackson Laboratory. All the mice used in this study were maintained in a pathogen-free animal barrier facility. The mice used in this study were housed under standardized conditions in a dedicated animal facility accredited by AAALAC, International, in compliance with institutional and ethical guidelines for the humane treatment of animals. The housing facility maintained a 12-h light/dark cycle, with lights on at 6:00 A.M. and off at 6:00 P.M. The ambient temperature was consistently maintained at $22 \pm 2$ °C, and humidity was kept within the range of 45% to 55%. All animals were housed in standard polypropylene cages with access to standard rodent chow and water ad libitum. The mice were acclimatized to these conditions for a minimum of 7 days before the commencement of any experimental procedures. All the experiments were initiated with mice of age 6 to 8 weeks. C42B EnzaRes Dox/shNME2 inducible cells were suspended at a concentration of 10 million cells/mL in 50% BD Matrigel– 50% PBS and 100 µL of this solution for a total of 1 million cells subcutaneously injected into the flanks of FVB mice. Mice were grouped and treatment was started when the tumor size reached around 150 to 180 mm3. The maximal tumor size burden permitted by Northwestern IACUC for mice is 2000 mm$^3$ and this was not exceeded for any of the animals involved in the study. Sex was not considered in the study design since these are prostate cancer models that only grow in male mice.

Enzalutamide was dissolved in 5% DMSO (Sigma-Aldrich #276855), 10% Solutol/Kolliphor HS15 (Sigma-Aldrich #42966) and 85% DPBS (Gibco #14190-144) at a concentration of 1.25 mg/mL for I.P. administration. Doxycycline hyclate (Sigma-Aldrich #D9891) was dissolved in drinking water at a concentration of 2 mg/mL and the solution supplemented with 5% sucrose w/v.

**Statistical analysis for in vitro and in vivo data.** All statistical tests in in vitro analyses were performed using one-tailed Welch t-test and implemented using the R *t.test* function[142]. For in vivo data, statistical comparison between groups was performed using two-way ANOVA, implemented in GraphPad Prism 9 software. Data are presented as mean ± standard error of the mean (S.E.M.) and statistical significance was defined by $P < 0.05$ (two-tailed). In figures, "*" indicates $P < 0.05$, "**" indicates $P < 0.01$, "***" indicates $P < 0.001$, "****" indicates $P < 0.0001$, and not significant "ns" indicates $P \geq 0.05$ for the indicated pairwise comparison.

**Reporting summary**
Further information on research design is available in the Nature Portfolio Reporting Summary linked to this article.

## Data availability
The mechanism-centric CRPC-specific network is available in Supplementary Data 2. Data supporting the findings of this study were obtained from (a detailed description of the datasets is available in Supplementary Data 1): (i) dbGaP: 1. Stand Up To Cancer East Coast Prostate Cancer cohort[32,33], RNA-sequencing data, phs000915.v2.p2. Access could be obtained through dbGap portal. 2. PROMOTE Prostate Cancer cohort[34], RNA-sequencing data, phs001141.v1.p1. Access could be obtained through dbGap portal. 3. Beltran et al.[68], RNA-sequencing data, phs000909.v1.p1. Access could be obtained through dbGap portal. 4. SU2C West Coast Prostate Cancer cohort[70,71], RNA-sequencing data, phs001648.v2.p1, downloaded from [https://portal.gdc.cancer.gov/projects/WCDT-MCRPC]. Access could be obtained through dbGap portal. (ii) cBioPortal: 1. Abida et al[33], RNA-sequencing data [https://github.com/cBioPortal/datahub/tree/master/public/prad_su2c_2019]. Data could be downloaded directly from cBio-Portal. (iii) GEO: 1. Kregel et al[53], microarray gene expression data, GSE78201. Data could be downloaded directly from GEO. 2. Kumar et al.[69], microarray gene expression data, GSE77930. Data could be downloaded directly from GEO. (iv) Broad Single Cell Portal: 1. He et al.[19], RNA-sequencing data [https://singlecell.broadinstitute.org/single_cell/study/SCP1244/transcriptional-mediators-of-treatment-resistance-in-lethal-prostate-cancer]. Data could be downloaded directly from Broad Single Cell Portal. (v) Manuscript Supplemental Information 1. Alumkal et al.[12], RNA-sequencing data: https://doi.org/10.1073/pnas.1922207117. Data could be obtained from the Supplementary Information in ref. 12. Hallmark and C2 pathway gene sets were obtained from the molecular signatures database (MSigDB 3.0) https://www.gsea-msigdb.org/gsea/msigdb. Source data are provided with this paper: experimental source data are provided as a Source Data file and computational source data are provided as a part of TR-2-PATH R software package (https://github.com/mitrofanova-lab/TR2PATH) under https://doi.org/10.5281/zenodo.10368948[158].

## Code availability
TR-2-PATH is released as an R software package in GitHub for community utilization (https://github.com/mitrofanova-lab/TR2PATH) under https://doi.org/10.5281/zenodo.10368948[158].

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

## Acknowledgements

We thank Nusrat J. Epsi, PostDoctoral fellow at Henry M. Jackson Foundation for Military Medicine, for her advice on estimating the activity level of transcriptional regulatory programs. We thank Carrie Esopenko, for sharing materials on partial least square regression analysis. Grant support includes New Jersey Commission on Cancer Research Pre-Doctoral Fellowship DCHS20PPC028 (S.P), *2021CIF-Rutgers-15, by the National Science Foundation (NSF) under Grant # 2127309 to the Computing Research Association for the CI-Fellows Post-Doctoral Fellowship Award* (M.W.C.), NIH NLM R01LM013236 (A.M., I.K.), ACS RSG-21-023-01-TBG (A.M., S.A., E.S., V.K.), NJCCR COCR21RBG00 (A.M., I.K.), NIH P50 CA180995 (S.A.A), NIH R01 CA257258 (S.A.A.), a Prostate Cancer Foundation (PCF) TACTICAL Award (S.A.A.), Polsky Urologic Cancer Institute (S.A.A.), and NIH F30 CA250196 (M.T.).

## Author contributions

S.P., M.I.T., V.K., S.A.A., and A.M. conceived and designed the study. A.M. and S.P. performed the computational and statistical analysis, prepared the figures, and wrote the paper. M.I.T. designed, performed, and analyzed in vitro and in vivo studies and prepared associated figures. C.Y.Y. performed the single-cell analysis. V.S. built the R software package. C.Y.Y. and V.S. contributed equally as co-second authors. M.W.C. and K.W. performed a quality check of the R software package. M.V.T. analyzed treatment information for CRPC patients. A.A.-S. and Sh.J. advised on the parallelization of RNA-seq data mapping. R.V. consulted on the VIF analysis. S.G. advised on comparison to other computational methods. J.S.P. advised on statistical methods utilized in the manuscript. S.Q. performed control in vitro experiments. F.C., Sh.G., and I.K. advised on the clinical utilization of the findings. E.S. supervised in vitro experiments. V.K. designed and performed in vitro analysis. S.A.A. conceived, designed, and analyzed in vitro and in vivo studies. All authors edited and approved the final manuscript.

## Competing interests

S.A.A. is a coinventor on patents covering the methods and assays to identify and characterize MYC inhibitors and derivatives (US11420957B2, "Substituted heterocycles as c-MYC targeting agents"). A.M., S.P., S.A.A. E.M.S., and V.K. are coinventors on patent applications covering the methods and assays to identify Enzalutamide-responsive tumors (PCT Application No. PCT/US2023/065533, "MYC program as a marker of response to Enzalutamide in prostate cancer"). A.M. and S.P. are coinventors on patent applications covering the computational method TR-2-PATH (U.S. Patent Application No. 18/297,470, "Identifying treatment response signatures"). The remaining authors declare no competing interests.
