## [Peer Review File · Nature Communications]

Reviewers' comments:

Reviewer #1 (Remarks to the Author); expert in clinical prostate cancer:

The study "Mechanism-centric network-based approach identifies NME2 and MYC programs as markers of resistance to Enzalutamide in CRPC patients" by Panja et al utilizes interesting approaches to identify upstream of MYC regulator associated with enzalutamide resistance in prostate cancer. Addressing the following comments may potentially further strengthen this very interesting study.

1. The authors demonstrate that high MYC levels are associated with enzalutamide resistance (Fig. 1C). They further show that high expression of both MYC and NME2 are associated with enzalutamide resistance (Fig. 5C). Are high levels of NME2 alone also associated with enzalutamide resistance? Are high levels of both MYC and NME2 better predictors for enzalutamide resistance when compared to each one alone?

2. The functional data in Figure 7 could be strengthened by the inclusion of in vivo data.

Reviewer #2 (Remarks to the Author); expert in prostate cancer gene regulation and enzalutamide resistance:

In the present study the authors investigated the mechanisms of enzalutamide (ENZ) resistance in CRPC patients by reconstructing a de novo mechanism-centric regulatory network that encodes relationships between molecular pathways and their upstream transcriptional regulatory programs (i.e., TR-2-PATH). Through this strategy, the authors identified MYC and NME2 programs as markers of resistance to ENZ in CRPC patients. Overall, this is an interesting study and the findings could be useful for development of new strategies to predict the ENZ resistance in prostate cancer patients in clinic. However, it appears a correlation study. The clinical relevance remains to be validated. Most importantly, further studies are required to define the regulatory mechanism between NME2 and MYC. Also, it remains unclear how MYC overexpression leads to ENZ resistance.

The rationale to select a small portion of CRPC patients from the cohort Abida et al. for the analysis of MYC association with ENZ resistance is not convincing. More information should be provided.

The authors indicated that there was no significant difference in Abiraterone-associated disease progression between patients from high MYC and normal/low MYC patients. Any explanation as to the specific effect of high MYC expression on resistance to ENZ, but not ABI?

It is unclear why the authors chose ONECUT2, ERG, and DLX1 genes as three markers to distinguish the high-MYC and high-NME2 group from the rest of the patients.

The authors sought to define the upstream regulator for high expression of MYC and they identified NME2 as the key molecule. However, the authors did not provide any evidence or hints as to how MYC expression can be influenced by NME2.

Through meta-analysis, the authors identified MYC as a potential driver of ENZ resistance in prostate cancer. However, they missed the opportunity to define the molecular basis underlying ENZ resistance driven by MYC.

Can the authors verify the positive correlation of MYC and NME2 protein expression in ENZ-resistant prostate cancer specimens in an independent cohort?

The MYC inhibitor MYCi975 has been reported previously and its anti-cancer effect is known.

The authors claimed that knockdown of NME2 further enhanced the inhibition of cell migration induced by the MYC inhibitor MYCi975. The data is not convincing, and no statistical analysis was performed. Even if this is the case, what is the molecular mechanism underlying the additive effect of NME2 knockdown and MYC inhibition?

While the in vitro additive anti-cancer effect of NME2 knockdown and MYC inhibition was relatively weak, are the results reproducible in vivo?

Reviewer #3 (Remarks to the Author); expert in systems biology and cancer:

In this paper Panja et al. used pathway and transcriptional regulatory program analysis to identify the role of MYC and NME2 as key regulators of Enzalutamide resistance in castration-resistant prostate cancer (CRPC). The authors used a wide variety of in vitro and in vivo (patient) datasets to identify MYC

and NME2 as potential resistance biomarkers, and computational tools to establish the importance of NME2 in MYC regulation. This kind of analysis can facilitate the use of systems biology tools in personalised medicine, thus can be interesting both for systems biology and cancer research / oncology readership. However, I have several questions regarding the used gene expression analysis and the experimental validation.

Major:

1) The authors show correlation between MYC activity and AR activity (SFig 1.) or different transcriptional regulatory programs, like NME2 (Fig. 3-4). However both pathway and TR activity is calculated using different “gene set enrichment” methods like ssGSEA and viper. To correctly interpret the functional relevance of pathway and TR activity correlations, the exact composition, and compositional similarity of the corresponding gene sets would be crucial - i.e. if the gene sets of 2 pathways / TR programs are highly overlapping, their functional similarity is not surprising. My opinion is that the composition of the used gene sets (at least the most important ones in Figure 1 and Figure 4) should be given as Supplementary data, and their compositional similarity, and its potentially confusing nature should be discussed.

2) The authors use the terminology “MYC pathway”, however it is not clear which pathway database (Hallmark, KEGG, REACTOME etc.) was used to define MYC pathway gene set. The same applies for AR “activity” - from which source did they define the AR gene set? Also, both MYC and AR are transcription factors, so they probably have transcriptional regulatory networks in MARINA. How do the composition of these gene sets correspond to the one’s used by the authors? By which criteria did they prefer one or the other?

3) The authors show that MYC and NME2 activity values are highly correlated. Has NME2 and MYC activity any potential advantage over simply MYC activity as biomarker?

4) Experimental validation: in Figure 7B, the authors show that “Interestingly, the colony formation ability was significantly reduced when MYCi975 and Enzalutamide were administered in combination on LNCaP-EnzaRes cells”. Does the reduction of colony formation in case of drug combination a consequence of drug synergy, or simply an additive effect? Also, the authors state: “Finally, to evaluate if silencing of NME2 could enhance the effect of MYC inhibition on cells’ 485 metastatic potential, we performed Boyden chamber-based in vitro migration assay using C42B-EnzaRes cells (see Methods), which demonstrated that the cell migration was further reduced when NME2 knockdown was added to MYCi975 administration (Fig. 7F).” Based on Figure 7F I do not see a strong decrease of migration using MYCi + siNME2 compared to MYCi or siNEM2 alone. Could the authors show some statistical tests to support their conclusion?

Minor

5) The authors use exaggerated expressions several times, like “striking concordance of their activity levels” and “striking reduction of the cell viability when treated with MYCi975” - I think more restrained expression would be more appropriate.

June 28, 2023

We would like to thank all three Reviewers for their careful consideration of our manuscript and insightful comments, which have greatly enhanced the significance and potential clinical relevance of our manuscript. We have carefully addressed all comments and outlined our responses below. Please note that **new in vivo, in vitro, and in silico data** are depicted in updated **Figure 7 and Supplementary Figures 7, 12, and 13 and also provided at the end of this document.**

REVIEWER #1 (Remarks to the Author); expert in clinical prostate cancer:

1. The study “Mechanism-centric network-based approach identifies NME2 and MYC programs as markers of resistance to Enzalutamide in CRPC patients” by Panja et al utilizes interesting approaches to identify upstream of MYC regulator associated with enzalutamide resistance in prostate cancer.

We would like to thank the Reviewer for their favorable evaluation of our manuscript and insightful comments.

2. The authors demonstrate that high MYC levels are associated with enzalutamide resistance (Fig. 1C). They further show that high expression of both MYC and NME2 are associated with enzalutamide resistance (Fig. 5C). Are high levels of NME2 alone also associated with enzalutamide resistance? Are high levels of both MYC and NME2 better predictors for enzalutamide resistance when compared to each one alone?

We would like to thank the Reviewer for this important point, which was not discussed fully in the previous manuscript version. Given a strong previously observed correlation between MYC and NME2 activities (Fig. 5C-D), we have further evaluated correlation of MYC and NME2 activities in three additional independent CRPC patient cohorts (Alumkal et al.¹ (n = 24), Beltran et al.² (n = 34) and Kumar et al.³ (n = 157), see new Supplementary Fig. 7, below), which confirmed significant correlation between MYC and NME2 programs (Pearson p-value = 3.53E-5, p-value = 5.13 E-11, and p-value < 2.2E-16, respectively).

Given MYC and NME2 consistent correlation in CRPC setting, we anticipate that both MYC pathway and NME2 TR program will be predictive of Enzalutamide (ENZ) resistance. In fact, when we compared ability of both MYC and NME2 to predict Enzalutamide resistance, to each one alone, we have observed significant performance of MYC pathway alone (log-rank p-value = 0.012) and NME2 alone (log-rank p-value = 0.0035) in predicting progression on Enzalutamide in patients – consistent with our new correlation studies. These results are also consistent with our findings that NME2 is upstream of MYC in Enzalutamide Resistant (EnzaRes) prostate cancer, as NME2 KD decreases MYC levels (Fig. 7H, below). We have added a statement explaining this to the “Results: Validation in clinical cohorts and Enzalutamide-specificity”.

3. The functional data in Figure 7 could be strengthened by the inclusion of in vivo data.

We have generated Enzalutamide Resistant (EnzaRes) cells with Dox-inducible KD of NME2. We show that NME2 KD (by Dox treatment) re-sensitized established Enzalutamide-resistant

prostate tumors to Enzalutamide treatment *in vivo* (Fig. 7I, Supplemental Fig. 12-13 - below).

REVIEWER 2 (Remarks to the Author); expert in prostate cancer gene regulation and enzalutamide resistance:

1. Overall, this is an interesting study and the findings could be useful for development of new strategies to predict the ENZ resistance in prostate cancer patients in clinic. However, it appears a correlation study. The clinical relevance remains to be validated. Most importantly, further studies are required to define the regulatory mechanism between NME2 and MYC. Also, it remains unclear how MYC overexpression leads to ENZ resistance.

We are thankful to the Reviewer for careful consideration of our manuscript and thoughtful comments and suggestions. We have now included additional functional *in vitro* and *in vivo* data on the relationship between NME2, MYC and Enzalutamide resistance. We show that NME2 is upstream of MYC using siRNA and newly-generated Dox-inducible shNME2 models (Fig. 7E, H below). We show *in vivo* that NME2 depletion re-sensitized Enzalutamide-resistant tumors to Enzalutamide (Fig 7I below).

The question of the mechanisms by which MYC overexpression leads to Enzalutamide (ENZ) resistance is an extensive one, that goes beyond the scope of this paper. Nevertheless, our group has recently shown that in ENZ resistant 22Rv1 cells MYC binds directly and regulates AR and the AR cofactor FOXA1⁴ [PMID: 35476451]. MYC inhibition with small-molecule MYC inhibitor MYCi975, reduced MYC binding to chromatin and expression of AR and FOXA1. Additionally, Labbe and colleagues have shown that MYC disrupts transcriptional pause release at androgen receptor targets which is associated with ENZ resistance⁵ [PMID: 35562350].

2. The rationale to select a small portion of CRPC patients from the cohort Abida et al. for the analysis of MYC association with ENZ resistance is not convincing. More information should be provided.

The reviewer brings an important point, which has been missed in our previous manuscript version. The rationale is available clinical information regarding exposure to ARSI and response to ARSI for these patients. Abida et al.⁶ cohort is CRPC cohort subjected to either Enzalutamide or Abiraterone. To examine Enzalutamide resistance specificity of MYC and NME2, we identified all ARSI-naïve patients later subjected to Enzalutamide (n = 22) or Abiraterone (n = 33), separately, from this cohort. To clarify this selection, we have added detailed explanations and now refer to the Enzalutamide subset of the cohort as *Enzalutamide-specific Abida et al cohort* and Abiraterone subset of the cohort as *Abiraterone-specific Abida et al cohort*, updated across Methods and Results.

3. The authors indicated that there was no significant difference in Abiraterone-associated disease progression between patients from high MYC and normal/low MYC patients. Any explanation as to the specific effect of high MYC expression on resistance to ENZ, but not ABI?

This is a question that intrigued us as well and is one that would require further investigation. We could speculate that it might potentially be related to the regulatory interactions between MYC

and AR (please note that our hypothesis is only speculative at this point). For example, our investigations of correlation between MYC pathway activity and AR expression and activity in Enzalutamide-specific and Abiraterone-specific Abida et al.⁶ cohorts (separately) demonstrated that the correlation between MYC pathway activity and AR expression/ AR activity in Enzalutamide-specific Abida et al. cohort is stronger (Spearman correlation rho = 0.482, p-value = 0.024; and Spearman correlation rho = 0.484, p-value = 0.023, for AR expression and AR activity respectively) compared to the same correlations in the Abiraterone-specific Abida et al cohort (Spearman correlation rho = 0.064, p-value = 0.719; and Spearman correlation rho = 0.071, p-value = 0.691, for AR expression and AR activity, respectively). However, extensive in-depth investigations would be needed to make a scientifically-grounded claim.

4. It is unclear why the authors chose ONECUT2, ERG, and DLX1 genes as three markers to distinguish the high-MYC and high-NME2 group from the rest of the patients.

We apologize if our language on the comparative analysis was not clear. We now carefully explain that alongside markers of ARSI and Enzalutamide resistance, we also evaluated 13 transcriptomic markers of PCa progression (identified from the current literature review), out of which three markers (ONECUT2, ERG, and DLX1) showed significant differential expression in the high-MYC and high-NME2 group. We have updated 2nd paragraph in “Results: Comparison to common markers of PCa aggressiveness and treatment response” to reflect this clarification.

5. The authors sought to define the upstream regulator for high expression of MYC and they identified NME2 as the key molecule. However, the authors did not provide any evidence or hints as to how MYC expression can be influenced by NME2.

We have experimentally addressed this issue in Enzalutamide Resistant (EnzaRes) PCa cells by demonstrating that NME2 siRNA or inducible shNME2 expression led to decreased MYC expression (Fig. 7E, H below).

6. Through meta-analysis, the authors identified MYC as a potential driver of ENZ resistance in prostate cancer. However, they missed the opportunity to define the molecular basis underlying ENZ resistance driven by MYC.

As mentioned in the response to the Reviewer’s comment #1 above, we do agree this is an important avenue for investigation but is somewhat beyond the scope of this paper. There have been a number of studies examining the relationship between MYC and response to ARSI. We recently showed that in ENZ resistant 22Rv1 cells MYC binds directly and regulates AR and the AR cofactor FOXA1⁴ [PMID: 35476451]. MYC inhibition with small-molecule MYC inhibitor MYCi975, reduced MYC binding to chromatin and expression of AR and FOXA1. Others have shown that MYC disrupts transcriptional pause release at androgen receptor targets which is associated with ENZ resistance⁵ [PMID: 35562350].

Furthermore, the new *in vivo* investigations now included (Fig. 7, Supplemental Fig. 12-13 below) support the notion that targeting the NME2/MYC axis can reverse Enzalutamide resistance.

7. Can the authors verify the positive correlation of MYC and NME2 protein expression in ENZ-resistant prostate cancer specimens in an independent cohort?

We would like to thank the Reviewer for this suggestion. Unfortunately, absence of a reliable NME2 antibody for immunohistochemistry has hampered our efforts to examine NME2 and MYC protein correlation in Enzalutamide Resistant (EnzaRes) patient samples. Multiple antibodies tested in our labs and the histopathology core at Rutgers University failed the appropriate quality controls, undermining confidence in their use.

However, to address this comment from the Reviewer, we assessed MYC and NME2 activity levels in three additional relevant CRPC cohorts, namely Alumkal et al.¹ (n = 24), Beltran et al.² (CRPC samples, n = 34), and Kumar et al.³ (n = 157). Such computationally estimated activity levels are based on the expression of MYC pathway genes and NME2 transcriptional targets and have been shown to be accurate predictions of the protein levels by our group and others⁷⁻¹⁰. This additional analysis demonstrated significant association between NME2 TR activity and MYC pathway activity in all three cohorts (Alumkal et al. Pearson r= 0.74, p-value = 3.53E-5; Beltran et al. Pearson r = 0.86, p-value = 5.13 E-11; Kumar et al. Pearson r = 0.65, p-value < 2.2E-16) and is now added to new Supplementary Figure 7 (below).

8. The MYC inhibitor MYCi975 has been reported previously and its anti-cancer effect is known.

We agree with the Reviewer. However, our study was not meant to report this inhibitor for the first time. We are rather using it as a validated MYC inhibitor in our studies. The main thrust of the current manuscript is the unbiased identification of a patient population that could benefit most from MYC (and NME2) inhibition. In particular, we propose that patients with activated levels of MYC and NME2 are at higher risk of developing resistance to Enzalutamide and would benefit from a combined therapeutic targeting along AR, MYC, and NME2 axes, as also demonstrated by our new *in vivo* investigations (updated Fig. 7 below).

9. The authors claimed that knockdown of NME2 further enhanced the inhibition of cell migration induced by the MYC inhibitor MYCi975. The data is not convincing, and no statistical analysis was performed. Even if this is the case, what is the molecular mechanism underlying the additive effect of NME2 knockdown and MYC inhibition?

We apologize for the error in the original migration quantification graph, unlike the images which were correct. This has now been fixed (Fig. 7F). NME2 KD suppresses MYC transcription at the level of MYC mRNA, while MYCi975 promotes MYC protein degradation by directly binding MYC without affecting MYC mRNA levels. The combination leads to a stronger suppression of MYC. [Note that MYCi975 is employed at a low dose below the IC50 in this cell line which is approx. 4 μ M].

10. While the in vitro additive anti-cancer effect of NME2 knockdown and MYC inhibition was relatively weak, are the results reproducible in vivo?

Yes. We generated Enzalutamide Resistant (EnzaRes) cells with Dox-inducible KD of NME2 to address this question. NME2 KD by Dox treatment reduced MYC levels and re-sensitized

established Enzalutamide-resistant prostate tumors to Enzalutamide treatment *in vivo* (Fig. 7I, Supplemental Fig. 13 below).

REVIEWER 3 (Remarks to the Author); expert in systems biology and cancer:

1. The authors used a wide variety of in vitro and in vivo (patient) datasets to identify MYC and NME2 as potential resistance biomarkers, and computational tools to establish the importance of NME2 in MYC regulation. This kind of analysis can facilitate the use of systems biology tools in personalised medicine, thus can be interesting both for systems biology and cancer research / oncology readership.

We would like to thank the Reviewer for favorable evaluation of the significance and innovation of our manuscript and for the thoughtful suggestions.

2. The authors show correlation between MYC activity and AR activity (SFig 1.) or different transcriptional regulatory programs, like NME2 (Fig. 3-4). However both pathway and TR activity is calculated using different “gene set enrichment” methods like ssGSEA and viper. To correctly interpret the functional relevance of pathway and TR activity correlations, the exact composition, and compositional similarity of the corresponding gene sets would be crucial - i.e. if the gene sets of 2 pathways / TR programs are highly overlapping, their functional similarity is not surprising. My opinion is that the composition of the used gene sets (at least the most important ones in Figure 1 and Figure 4) should be given as Supplementary data, and their compositional similarity, and its potentially confusing nature should be discussed.

This is a very important point and great suggestion. The Reviewer is absolutely correct that if such target genes were highly overlapping, the functional similarity of MYC, NME2, and AR would not be surprising. In fact, before performing the analysis we ensured that the overlap of targets/members in these sets was not a factor in the analysis, yet we did not provide this information in the manuscript before – in it now included.

In particular, MYC pathway (HALLMARK_MYC_TARGETS_V2) is comprised of 58 genes and the transcriptional regulon of AR is comprised of 231 genes (both of which are now provided in Supplementary Table 2). These two gene sets are highly non-overlapping (common genes = 2, Jaccard similarity index = 0.007 out of 1).

At the same time, MYC pathway is comprised of 58 genes and the transcriptional regulon of NME2 is comprised of 412 genes (both of which are now provided in Supplementary Table 6). These two gene sets are highly non-overlapping (common genes = 7, Jaccard similarity index = 0.01 out of 1).

We have now provided the list of all targets in Supplementary Table 2 and Supplementary Table 6 and have added the corresponding statistics and discussion to sections “Results: Increased MYC activity is specific to the risk of Enzalutamide resistance in CRPC patients” and “Results: Network Mining II: Prioritization of upstream regulatory programs” sections.

3. The authors use the terminology “MYC pathway”, however it is not clear which pathway database (Hallmark, KEGG, REACTOME etc.) was used to define MYC pathway gene set. The same applies for AR “activity” - from which source did they define the AR gene set? Also, both MYC and AR are transcription factors, so they probably have transcriptional regulatory networks in MARINA. How do the composition of these gene sets correspond to the one’s used by the authors? By which criteria did they prefer one or the other?

We thank the Reviewer for this important comment and suggestion, which we overlooked. HALLMARK_MYC_TARGETS_V2 pathway (n = 58) from Hallmark collection showed the most significant enrichment in the Enzalutamide-resistant phenotype and thus has been utilized for our analysis. This information is now included in “Results: Increased MYC activity is specific to the risk of Enzalutamide resistance in CRPC patients” (2nd paragraph).

With respect to estimation of the transcriptional regulatory activity (such as for AR, NME2, and all other transcriptional regulators) for consistency we utilized MARINA and VIPER on the prostate-cancer specific transcriptional regulatory networks (interactome), reconstructed by our group previously in Aytes et al.¹¹. We now highlight this information in the main body of the manuscript.

In addition, we have provided the list of all MYC pathway genes, and AR and NME2 transcriptional targets and their overlaps in Supplementary Table 2 and Supplementary Table 6.

4. The authors show that MYC and NME2 activity values are highly correlated. Has NME2 and MYC activity any potential advantage over simply MYC activity as biomarker?

We would like to thank the Reviewer for this important point, which was not discussed fully in the previous manuscript version. While utilization of MYC pathway alone could predict Enzalutamide resistance (Fig 1C, log-rank p-value = 0.012, adjusted HR = 4.39), utilizing both MYC and NME2 provides a better overall predictive model (Fig 5C, log-rank p-value = 0.0035, adjusted HR = 5.28). However, in our opinion, the strongest advantage of using NME2 alongside MYC lies in their combined therapeutic targeting. In this revised manuscript, we have demonstrated that targeting NME2 and MYC together produces the strongest response both *in vitro* and *in vivo* experimental systems, when compared to targeting either of them alone (updated Fig. 7 below).

5. Experimental validation: in Figure 7B, the authors show that “Interestingly, the colony formation ability was significantly reduced when MYCi975 and Enzalutamide were administered in combination on LNCaP-EnzaRes cells”. Does the reduction of colony formation in case of drug combination a consequence of drug synergy, or simply an additive effect?

These colony forming assay data shown here with a single dose each for each drug are not robust for calling out formal synergy, which is why we did not state that. MYCi975 synergy with Enzalutamide has been formally demonstrated with BLISS analysis by our group in another context though in 22Rv1, C42B, and LNCaP cells [Fig. 6C in Holmes et al.⁴ PMID: 35476451].

6. Also, the authors state: “Finally, to evaluate if silencing of NME2 could enhance the effect of MYC inhibition on cells’ 485 metastatic potential, we performed Boyden chamber-based in vitro migration assay using C42B-EnzaRes cells (see Methods), which demonstrated that the cell migration was

further reduced when NME2 knockdown was added to MYCi975 administration (Fig. 7F).” Based on Figure 7F I do not see a strong decrease of migration using MYCi + siNME2 compared to MYCi or siNEM2 alone. Could the authors show some statistical tests to support their conclusion?

The quantification graphs in this figure contained an error of scaling which has now been corrected. The original images are accurate, remain unchanged and reflect the difference stated in the results paragraph. We have now also performed 2-way ANOVA on the quantified results and have updated the graph with the appropriate p-value statistical significance symbols.

7. The authors use exaggerated expressions several times, like “striking concordance of their activity levels” and “striking reduction of the cell viability when treated with MYCi975” - I think more restrained expression would be more appropriate.

We agree with the Reviewer’s comment and have removed expressions such as “striking” throughout the text.

We appreciate all of the Reviewers’ comments and suggestions, which have made our manuscript stronger and highlighted its significance and potential clinical relevance.

- 1 Alumkal, J. J. *et al.* Transcriptional profiling identifies an androgen receptor activity-low, stemness program associated with enzalutamide resistance. *Proceedings of the National Academy of Sciences* **117**, 12315 (2020). <https://doi.org:10.1073/pnas.1922207117>
- 2 Beltran, H. *et al.* Divergent clonal evolution of castration-resistant neuroendocrine prostate cancer. *Nat Med* **22**, 298-305 (2016). <https://doi.org:10.1038/nm.4045>
<http://www.nature.com/nm/journal/v22/n3/abs/nm.4045.html#supplementary-information>
- 3 Kumar, A. *et al.* Substantial interindividual and limited intraindividual genomic diversity among tumors from men with metastatic prostate cancer. *Nat Med* **22**, 369-378 (2016). <https://doi.org:10.1038/nm.4053>
- 4 Holmes, A. G. *et al.* A MYC inhibitor selectively alters the MYC and MAX cisomes and modulates the epigenomic landscape to regulate target gene expression. *Science Advances* **8**, eabh3635 (2022). <https://doi.org:doi:10.1126/sciadv.abh3635>

- 5 Qiu, X. *et al.* MYC drives aggressive prostate cancer by disrupting transcriptional pause release at androgen receptor targets. *Nat Commun* **13**, 2559 (2022). <https://doi.org:10.1038/s41467-022-30257-z>
- 6 Abida, W. *et al.* Genomic correlates of clinical outcome in advanced prostate cancer. *Proceedings of the National Academy of Sciences* **116**, 11428 (2019). <https://doi.org:10.1073/pnas.1902651116>
- 7 Arriaga, J. M. *et al.* A MYC and RAS co-activation signature in localized prostate cancer drives bone metastasis and castration resistance. *Nature Cancer* **1**, 1082-1096 (2020). <https://doi.org:10.1038/s43018-020-00125-0>
- 8 Epsi, N. J., Panja, S., Pine, S. R. & Mitrofanova, A. pathCHEMO, a generalizable computational framework uncovers molecular pathways of chemoresistance in lung adenocarcinoma. *Communications Biology* **2**, 334 (2019). <https://doi.org:10.1038/s42003-019-0572-6>
- 9 Alvarez, M. J. *et al.* Functional characterization of somatic mutations in cancer using network-based inference of protein activity. *Nature genetics* **48**, 838 (2016).
- 10 Lefebvre, C. *et al.* A human B-cell interactome identifies MYB and FOXM1 as master regulators of proliferation in germinal centers. *Mol Syst Biol* **6**, 377-377 (2010). <https://doi.org:10.1038/msb.2010.31>
- 11 Aytes, A. *et al.* Cross-species regulatory network analysis identifies a synergistic interaction between FOXM1 and CENPF that drives prostate cancer malignancy. *Cancer Cell* **25**, 638-651 (2014). <https://doi.org:10.1016/j.ccr.2014.03.017>

FIGURE 7:

Fig. 7: MYC targeting is beneficial in Enzalutamide-resistant conditions. (A) Drug sensitivity curves of Enzalutamide-naïve, or Enzalutamide-resistant (EnzaRes) C42B cells treated with MYCi975. (B) Colony formation assay using Enzalutamide-resistant (EnzaRes) C42B cells in Intact (i.e., treated with DMSO), treated with Enzalutamide (10 μM), MYCi975 (2 μM), or a combination of Enzalutamide+MYCi975 (10 μM +2 μM). Cells were grown in the presence of respective drugs. Representative images are shown. P-value was estimated using a one-tailed Welch t-test. * p-value < 0.05, *** p-value < 0.001 (C) Boyden chamber-based *in vitro* migration assay using Enzalutamide-resistant (EnzaRes) C42B cells in Intact (i.e., treated with DMSO), treated with Enzalutamide (10 μM), MYCi975 (2 μM), or a combination of Enzalutamide+MYCi975 (10 μM +2 μM). Bars represent the quantification of Crystal Violet trapped by migrated cells. P-value was estimated using a one-tailed Welch t-test. * p-value < 0.05, ** p-value < 0.01 (D) Expression of NME2 in Intact and

Enzalutamide-resistant (EnzaRes) C42B cells, using qRT-PCR. P-value was estimated using the one-tailed Welch t-test. *** p-value ≤ 0.001 **(E)** Two different siRNAs targeting NME2 were used to downregulate NME2 (left panel) and its effect on MYC expression using qRT-PCR is shown (right panel). P-value was estimated using one-tailed Welch t-test * p-value < 0.05 **(F)** Boyden chamber-based in vitro migration assay using Enzalutamide-resistance (EnzaRes) C42B cells treated with DMSO or MYCi975 (2 μ M) with or without knockdown of NME2. Bars represent the cell count quantification of Crystal Violet trapped by migrated cells, normalized to control (DMSO with siScramble). P-value was estimated using 2-way ANOVA, * p-value < 0.05 , ** p-value < 0.01 , *** p-value < 0.001 , **** p-value < 0.0001 . **(G)** Comparison of anti-proliferative effects of Enzalutamide on C42B EnzaRes cells that have NME2 knocked down via CRISPR (sgNME2) versus C42B EnzaRes which have received non-targeting sgRNA control (sgNT). **(H)** Western blot of NME2, MYC and GAPDH protein levels in C42B shNME2 EnzaRes cells treated with Doxycycline for 96 hours at the indicated concentrations. **(I)** Schematic representation and tumor volumes for mice bearing established C42B EnzaRes shNME2 xenografts treated with vehicle (n = 7 mice), Enzalutamide (10 mg/kg QD i.p) and/or Doxycycline (2 mg/mL in drinking water) (n = 6 mice for Dox, Enza, Enza + Dox arms) for 24 days. Statistical significance performed via two-way ANOVA. * p-value < 0.05 , *** p-value ≤ 0.001 , **** p-value ≤ 0.0001 .

SUPPLEMENTARY FIGURE 7:

Supplementary Fig. 7 (Related to Fig. 5). Comparative analysis between NME2 TR and MYC pathway activities demonstrates their correlation across multiple CRPC cohorts. Pearson correlation analysis between NME2 TR activity and MYC pathway activity in Alumkal et al., Beltran et al. (CRPC samples), and Kumar et al. CRPC cohorts. Pearson r and p-values are indicated.

SUPPLEMENTARY FIGURE 12:

Supplementary Fig. 12 (Related to Fig. 7). Validation of NME2 CRISPR knockout in C42B- EnzaRes cells. Western blot of NME2 and GAPDH protein levels in C4-2B EnzaRes cells after performing CRISPR KO with two different sgRNAs targeting NME2, inserted in the pLentiCRISPR-v2 lentiviral plasmid.

SUPPLEMENTARY FIGURE 13:

Supplementary Fig. 13 (Related to Fig. 7). Individual tumor growth curves and mouse body weight plots for in vivo study. (A) Tumor growth curves over duration of treatment for individual mice from each treatment arm from the study shown in Fig. 7I (B) Mouse average body weight over the duration of treatment from the study shown in Fig. 7I.

REVIEWER COMMENTS

Reviewer #1 (Remarks to the Author):

The revised manuscript "Mechanism-centric network-based approach identifies NME2 and MYC programs as markers of resistance to Enzalutamide in CRPC patients" by Panja et al utilizes interesting approaches to identify upstream of MYC regulator associated with enzalutamide resistance in prostate cancer. All prior comments are very well addressed and new data to support the study has been included. The study is recommended for publication in Nature Communication by this Reviewer.

Reviewer #3 (Remarks to the Author):

The authors answered all my concerns, and the manuscript nicely improved.

Reviewer #4 (Remarks to the Author): Expert in prostate cancer gene regulation, therapy and enzalutamide resistance

The study titled 'Mechanism-centric network-based approach identifies NME2 and MYC programs as markers of resistance to Enzalutamide in CRPC patients' is a well written evaluation of the underlying transcriptional drivers of Myc expression in prostate cancer. The authors have done a credible job of addressing the prior reviewers comments. The results identify a potential new pathway for targeting in enzalutamide resistance. However, additional issues must be addressed prior to acceptance of this manuscript.

1. The differences in myc expression between the enzalutamide-responsive and enzalutamide-resistant abida cohort and abiraterone abida cohorts in Fig 1. remains unexplained. Given that the #s of patient samples used to make this claim is small n=22 and n=33, the authors should clearly state that this is a very preliminary finding and that needs to be validated in larger cohorts before using this finding. I would encourage the authors to simply state that in the abiraterone treated cohort, a similar correlation with myc expression was not noted. The statements on line 160-163 that the increased activity of Myc pathway is specific for enzalutamide cannot be made from this initial observation, given the small numbers of patients and lack of validation in other cohorts. Rather, simply state that the correlation was noted in patient samples treated with enzalutamide.

2. if the abiraterone treatment did not stratify for myc pathways expression, then why was the data from these patients included in the SU2C cohort used for analyses in Fig.2. Why wasn't that analysis only performed with patients treated with enzalutamide? Ideally, the authors should repeat the analyses from the samples of patients only treated with enzalutamide and confirm that their findings remain relevant.

3. The authors have described well their innovative approach to identify TR programs upstream of myc using cell line data in Fig 3 and 4.

4. The data in Fig. 5 is anecdotal and supportive of the paper and should be framed as such. The small numbers of patients at risk in the data from Figs. 5C and 5D make the data uninterpretable and statistically meaningless. The comparison of 3 vs 7 patients at 10 months and 1 vs 4 patients at 20m in Fig 5 C and 2 vs 9 patients at 5m and 1 vs 1 patient at 10m in Fig 5D are best suggestive and not definitive. Further stratification of the small number of patients from Supp Fig 8A-F should be deleted. Inclusion of the SU2C West coast data in Fig. 5D is confusing, as it contains data from both abiraterone and enzalutamide treated patients. IT would be useful to understand what proportion of those 83 patients had treatment with enzalutamide or abiraterone. An alternative dataset if available should be considered. In addition, the conclusions from this Figure should be toned down... the figure 5 legend should read "activity of myc and NME2 are associated with poor response to Enzalutamide".

5. The validation studies in Figure 7 would benefit from some additional characterization, including inclusion of a positive control for myc activity (myc protein levels after treatment with MYCi95 at doses used in Fig. 7A-C)), additional comparisons of NME2 levels in Enza-res LNCAP cells (Fig. 7D, inclusion of both knockdowns from supp fig 12 in Fig 7G and evaluation of myc levels in supp fig. 12.

6. The authors have partially addressed the reviewers comments with their in vivo studies. Given the modest knockdown of NME2 with shRNA and better knockdown with CRISPR, why wasn't the in vivo comparison performed between the C4-2b enzaRes cells without and with NME2 knockdown? Perhaps if these cells with NME2 knockdown form tumors, their response to enzalutamide may be more dramatic?

Overall, this is a highly innovative computational approach to identify TRs upstream of myc.

September 26, 2023

We would like to thank all four Reviewers for their careful consideration of our manuscript and insightful comments, which have greatly enhanced our manuscript. We have carefully addressed additional comments from Reviewer 4 and outlined our responses below. Please note that in addition to text changes, **new in vitro data** are depicted in updated **Figure 7G and new Supplementary Figures 10C and 11 and also provided at the end of this document.**

REVIEWER #4 (Remarks to the Author): expert in prostate cancer gene regulation, therapy and enzalutamide resistance:

The study titled 'Mechanism-centric network-based approach identifies NME2 and MYC programs as markers of resistance to Enzalutamide in CRPC patients' is a well written evaluation of the underlying transcriptional drivers of Myc expression in prostate cancer. The authors have done a credible job of addressing the prior reviewers comments. The results identify a potential new pathway for targeting in enzalutamide resistance. However, additional issues must be addressed prior to acceptance of this manuscript.

We would like to thank the Reviewer for their favorable evaluation of our manuscript and thoughtful and insightful comments. We have carefully addressed the Reviewer's comments in the manuscript and below.

1. The differences in myc expression between the enzalutamide-responsive and enzalutamide-resistant abida cohort and abiraterone abida cohorts in Fig 1. remains unexplained. Given that the #s of patient samples used to make this claim is small n=22 and n=33, the authors should clearly state that this is a very preliminary finding and that needs to be validated in larger cohorts before using this finding. I would encourage the authors to simply state that in the abiraterone treated cohort, a similar correlation with myc expression was not noted. The statements on line 160-163 that the increased activity of Myc pathway is specific for enzalutamide cannot be made from this initial observation, given the small numbers of patients and lack of validation in other cohorts. Rather, simply state that the correlation was noted in patient samples treated with enzalutamide.

We agree and have toned down comparison to Abiraterone (stating that a similar correlation between MYC and Abiraterone response was not observed) and indicated that validation in larger cohorts would be valuable for future clinical use. These changes are now reflected across the manuscript, in particular in lines 101-102, 113-114, 151-153, 159-160, 163-166, 300, 374-376, and 390-393 and below.

In particular,

Lines 101-102: "Further, we evaluated MYC and NME2 partnership in response to Abiraterone (a second-generation AR signaling inhibitor of androgen biosynthesis) and demonstrated that it is not associated to that the risk of resistance to Abiraterone is indeed specific to Enzalutamide."

Lines 113-114: "Increased MYC activity is specific to associated with the risk of Enzalutamide resistance in CRPC patients."

Lines 151-153: “Interestingly, in Abiraterone-specific Abida et al cohort, we did not observe a To further evaluate if this finding was specific to Enzalutamide treatment, we tested if correlation of MYC pathway activity with could predict treatment response to Abiraterone response.”

Lines 159-160: “Contrary to the results obtained from Enzalutamide associated disease progression, We identified no significant difference in Abiraterone-associated disease progression between patients from high MYC and normal/low MYC groups.”

Lines 163-166: “Taken together, these analyses demonstrate that increased activity of MYC pathway is indicative of a higher risk of resistance specifically to Enzalutamide and could potentially serve as a marker to stratify patients for their risk of developing resistance to Enzalutamide and would benefit from validation in larger patient cohorts for future clinical use.”

Line 300: “Validation in clinical cohorts and Enzalutamide specificity.”

Lines 374-376: “To further investigate and confirm that the activation of evaluate association of NME2 TR and MYC pathway with response to Abiraterone could specifically predict Enzalutamide failure (and not, for example, Abiraterone failure)...”

Lines 390-393: “Taken together, these analyses suggest, demonstrating that partnership between NME2 TR and MYC pathway is associated with specifically indicative of the risk of developing resistance response to Enzalutamide and not Abiraterone.”

2. If the abiraterone treatment did not stratify for myc pathways expression, then why was the data from these patients included in the SU2C cohort used for analyses in Fig.2. Why wasn't that analysis only performed with patients treated with enzalutamide? Ideally, the authors should repeat the analyses from the samples of patients only treated with enzalutamide and confirm that their findings remain relevant.

We thank the Reviewer for bringing this important point to our attention. We apologize that our choice of the SU2C cohort for network reconstruction was not clear from our previous paper version. We now added explanation to the manuscript that the overall objective of the mechanism-centric network reconstruction was to build a network that is specific for CRPC, but not limited to any CRPC-related phenotype (treatment, metastatic site etc.). This network would provide a comprehensive resource to the prostate cancer community and could be used to answer multiple CRPC-related questions (e.g., primary and secondary treatment resistance, metastases to different sites etc.). At the same time, the subsequent step (network mining) provides an opportunity to query this network with a specific CRPC-related question (in our case, resistance to Enzalutamide). We have now included additional explanation and rationale for this network reconstruction using diverse CRPC samples from SU2C cohort in the manuscript in lines 176-180 and below.

In particular,

Lines 176-180: “Our objective was to reconstruct a mechanism-centric regulatory network that would capture a wide-array of CRPC-specific phenotypes, constituting a valuable resource for the community and allowing effective utilization across different CRPC-related questions (e.g., primary and secondary therapeutic resistance, metastases to different sites etc.)”

3. The authors have described well their innovative approach to identify TR programs upstream of myc using cell line data in Fig 3 and 4.

We thank the Reviewer for this encouraging comment.

4. The data in Fig. 5 is anecdotal and supportive of the paper and should be framed as such. The small numbers of patients at risk in the data from Figs. 5C and 5D make the data uninterpretable and statistically meaningless. The comparison of 3 vs 7 patients at 10 months and 1 vs 4 patients at 20m in Fig 5 C and 2 vs 9 patients at 5m and 1 vs 1 patient at 10m in Fig 5D are best suggestive and not definitive. Further stratification of the small number of patients from Supp Fig 8A-F should be deleted. Inclusion of the SU2C West coast data in Fig. 5D is confusing, as it contains data from both abiraterone and enzalutamide treated patients. IT would be useful to understand what proportion of those 83 patients had treatment with enzalutamide or abiraterone. An alternative dataset if available should be considered. In addition, the conclusions from this Figure should be toned down... the figure 5 legend should read "activity of myc and NME2 are associated with poor response to Enzalutamide".

We agree with the Reviewer and removed Supplementary Fig 8 and removed related text (lines 349 - 354), changed Figure 5 legend (lines 1717-1718), and toned down conclusions in the "Validation in clinical cohorts" section (lines 344-349 and 390-393). With respect to SU2C West Coast cohort, we have combined clinical information from Quigley et al.² and Westbrook et al.³ publications that utilize SU2C West Coast cohort, which demonstrated that 67% of patients in this cohort were pre- or post-treated with Enzalutamide (either Enzalutamide alone or in combination with Abiraterone). We apologize that this information was not included in the previous paper version and have now included this important detail in the manuscript lines 355-358. Even though we cannot directly separate Enzalutamide-treated patients from Abiraterone -treated patients in this cohort, given the fact that 67% of patients were subjected to Enzalutamide, this dataset constitutes a valuable resource for our investigations. We have performed a comprehensive database search, but were not able to identify additional datasets to be included.

In particular,

Lines 1717-1718: "Activities of MYC and NME2 are associated with poor response to Enzalutamide"

Lines 344-349: "A comparison of these groups using Kaplan-Meier survival analysis and Cox proportional hazards model analysis, demonstrated a significant difference in their Enzalutamide-associated disease progression (Fig. 5C, right, log-rank p-value = 0.0035, adjusted HR = 5.28, CI = 1.58 - 18.38, see Methods),, suggesting that the high activity levels of NME2 TR and MYC pathway can be utilized to predict CRPC patients who are at a higher risk of developing resistance to Enzalutamide, prior to therapy administration. "

Lines 390-393: "Taken together, these analyses suggest, demonstrating that partnership between NME2 TR and MYC pathway is associated with specifically indicative of the risk of developing resistance resistance toto Enzalutamide and not Abiraterone."

Lines 355-358: "To extend our validation studies to an additional patient cohort, we utilized SU2C West Coast cohort^{70,71} (see Methods, Supplementary Table 1, n = 83) which comprises of CRPC patients that were subjected to Enzalutamide and/or Abiraterone (~67% of patients in this cohort were pre- or post-treated with Enzalutamide, either alone or in combination with Abiraterone...)"

5. The validation studies in Figure 7 would benefit from some additional characterization, including inclusion of a positive control for myc activity (myc protein levels after treatment with MYCi95 at doses

used in Fig. 7A-C)), additional comparisons of NME2 levels in Enza-res LNCAP cells (Fig. 7D, inclusion of both knockdowns from supp fig 12 in Fig 7G and evaluation of myc levels in supp fig. 12.

We thank the Reviewer for these important points, which were not included in the previous manuscript version. Positive control for MYC protein level after treatment with MYCi975 in C42B-EnzaRes cells is now included in **Supplementary Fig. 11** and lines 501-503 (note that the order of supplementary figures has changed, compared to the initial submission). NME2 levels in LNCaP-EnzaRes cells vs control are included in **Supplementary Fig. 10C** and lines 504-508. Both knockdowns from Supplementary Fig. 12 are now included in **Figure 7G** and figure legend lines (we performed new IC₅₀ assays in triplicate).

In particular,

Lines 501-503:” Positive control for MYC protein level after treatment with MYCi975 in C42B-EnzaRes cells was performed after 24 hours of treatment (Supplementary Fig. 11)”

Lines 504-508: “SubsequentlyNext, to evaluate the effect of NME2 silencing on MYC expression, we first evaluated the expression of NME2 in LNCaP/C42B-EnzaRes cells, compared to wild-type LNCaP/C42B cells, which showed elevated levels of NME2 in both LNCaP-EnzaRes cells (Supplementary Fig 10C, one-tailed Welch t-test p-value = 0.00188, see Methods) and C42B-EnzaRes cells (Fig. 7D, one-tailed Welch t-test p-value = 0.0005, see Methods).”

Lines 1786-1788: “(G) Comparison of anti-proliferative effects of Enzalutamide on C42B EnzaRes cells that have NME2 knocked down via CRISPR (sgNME2_V1 and sgNME2_V2) versus C42B EnzaRes which have received non-targeting sgRNA control (sgNT).”

6. The authors have partially addressed the reviewers comments with their *in vivo* studies. Given the modest knockdown of NME2 with shRNA and better knockdown with CRISPR, why wasn't the *in vivo* comparison performed between the C4-2b enzaRes cells without and with NME2 knockdown? Perhaps if these cells with NME2 knockdown form tumors, their response to enzalutamide may be more dramatic?

In terms of shRNA vs CRISPR choice for our *in vivo* study, we reasoned that the shRNA model is a better mimic of real-life clinical scenario. The C42B-EnzaRes shRNA cells can establish the tumor in the context of high NME2 levels, as opposed to the CRISPR cells where the tumor would be established in the context of almost complete NME2 knockdown. Furthermore, more limited NME2 knockdown is representative of a putative therapeutic arm in patients where it is reasonable to expect that regardless of modality (direct or indirect NME2 inhibitor, for example) levels of NME2 would not be completely suppressed. Finally, demonstration of restoration of Enza sensitivity in the context of residual NME2 levels suggests that this therapeutic approach is promising even under more realistic conditions of partial NME2 suppression.

Overall, this is a highly innovative computational approach to identify TRs upstream of myc.

We highly appreciate Reviewer’s positive evaluation of our manuscript and constructive and thoughtful comments and suggestions.

We appreciate all of the Reviewers' comments and suggestions, which have made our manuscript stronger and highlighted its significance and innovation.

1. Abida W, Cyrta J, Heller G, Prandi D, Armenia J, Coleman I, Cieslik M, Benelli M, Robinson D, Van Allen EM, Sboner A, Fedrizzi T, Mosquera JM, Robinson BD, De Sarkar N, Kunju LP, Tomlins S, Wu YM, Nava Rodrigues D, Loda M, Gopalan A, Reuter VE, Pritchard CC, Mateo J, Bianchini D, Miranda S, Carreira S, Rescigno P, Filipenko J, Vinson J, Montgomery RB, Beltran H, Heath EI, Scher HI, Kantoff PW, Taplin M-E, Schultz N, deBono JS, Demichelis F, Nelson PS, Rubin MA, Chinnaiyan AM, Sawyers CL. Genomic correlates of clinical outcome in advanced prostate cancer. *Proceedings of the National Academy of Sciences*. 2019;116(23):11428. doi: 10.1073/pnas.1902651116.
2. Quigley DA, Dang HX, Zhao SG, Lloyd P, Aggarwal R, Alumkal JJ, Foye A, Kothari V, Perry MD, Bailey AM, Playdle D, Barnard TJ, Zhang L, Zhang J, Youngren JF, Cieslik MP, Parolia A, Beer TM, Thomas G, Chi KN, Gleave M, Lack NA, Zoubeidi A, Reiter RE, Rettig MB, Witte O, Ryan CJ, Fong L, Kim W, Friedlander T, Chou J, Li H, Das R, Li H, Moussavi-Baygi R, Goodarzi H, Gilbert LA, Lara PN, Jr., Evans CP, Goldstein TC, Stuart JM, Tomlins SA, Spratt DE, Cheetham RK, Cheng DT, Farh K, Gehring JS, Hakenberg J, Liao A, Febbo PG, Shon J, Sickler B, Batzoglou S, Knudsen KE, He HH, Huang J, Wyatt AW, Dehm SM, Ashworth A, Chinnaiyan AM, Maher CA, Small EJ, Feng FY. Genomic Hallmarks and Structural Variation in Metastatic Prostate Cancer. *Cell*. 2018;174(3):75869.e9. Epub 2018/07/24. doi: 10.1016/j.cell.2018.06.039. PubMed PMID: 30033370; PMCID: PMC6425931.
3. Westbrook TC, Guan X, Rodansky E, Flores D, Liu CJ, Udager AM, Patel RA, Haffner MC, Hu Y-M, Sun D, Beer TM, Foye A, Aggarwal R, Quigley DA, Youngren JF, Ryan CJ, Gleave M, Wang Y, Huang J, Coleman I, Morrissey C, Nelson PS, Evans CP, Lara P, Reiter RE, Witte O, Rettig M, Wong CK, Weinstein AS, Uzunangelov V, Stuart JM, Thomas GV, Feng FY, Small EJ, Yates JA, Xia Z, Alumkal JJ. Transcriptional profiling of matched patient biopsies clarifies molecular determinants of enzalutamide-induced lineage plasticity. *Nature Communications*. 2022;13(1):5345. doi: 10.1038/s41467-022-32701-6.

SUPPLEMENTARY FIGURE 10:

Supplementary Fig. 10 (Related to Fig. 7). MYC inhibition reduces viability and colony formation in Enzalutamide-resistant conditions. (A) Drug response curves of Enzalutamide-naïve LNCaP or Enzalutamide-resistant LNCaP cells treated with Enzalutamide and/or MYCi975. **(B)** Colony formation assay using Enzalutamide resistant LNCaP cells (LNCaP-EnzaRes) in Intact (treated with DMSO), treated with Enza (10μM), MYCi975 (2μM) or a combination of Enza+MYCi975 (10μM+2μM). Cells were grown in the presence of respective drugs. Bars represent quantification of Crystal Violet trapped by migrated cells. P-value is estimated utilizing one-tailed Welch t-test. * p-value < 0.05, *** p-value ≤ 0.001. **(C)** Expression of NME2 in Intact (DMSO) and Enzalutamide-resistant (EnzaRes) LNCaP cells, using qRT-PCR. P-value is estimated using one-tailed Welch t-test. ** p-value ≤ 0.01 .

SUPPLEMENTARY FIGURE 11:

Supplementary Fig. 11 (Related to Fig. 7). MYCi degrades MYC in C42B-EnzaRes cells. Western blot of c-MYC and actin protein levels in C42B-EnzaRes cells after treatment with MYCi975 for 24 hours at indicated concentrations.

FIGURE 7:

Fig. 7: MYC targeting is beneficial in Enzalutamide-resistant conditions. (A) Drug sensitivity curves of Enzalutamide-naïve, or Enzalutamide-resistant (EnzaRes) C42B cells treated with MYCi975. (B) Colony formation assay using Enzalutamide-resistant (EnzaRes) C42B cells in Intact (i.e., treated with DMSO), treated with Enzalutamide (10μM), MYCi975 (2μM), or a combination of Enzalutamide+MYCi975 (10μM+2μM). Cells were grown in the presence of respective drugs. Representative images are shown. P-value was estimated using a one-tailed Welch t-test. * p-value < 0.05, *** p-value < 0.001 (C) Boyden chamber-based *in vitro* migration assay using Enzalutamide-resistant (EnzaRes) C42B cells in Intact (i.e., treated with DMSO), treated with Enzalutamide (10μM), MYCi975 (2μM), or a combination of Enzalutamide+MYCi975 (10μM+2μM). Bars represent the quantification of Crystal Violet trapped by migrated cells. P-value was estimated using a one-tailed Welch t-test. * p-value < 0.05, ** p-value < 0.01 (D) Expression of NME2 in Intact and Enzalutamide-resistant (EnzaRes) C42B cells, using qRT-PCR. P-value was estimated using the one-tailed

Welch t-test. *** p-value ≤ 0.001 **(E)** Two different siRNAs targeting NME2 were used to downregulate NME2 (left panel) and its effect on MYC expression using qRT-PCR is shown (right panel). P-value was estimated using one-tailed Welch t-test * p-value < 0.05 **(F)** Boyden chamber-based in vitro migration assay using Enzalutamide-resistance (EnzaRes) C42B cells treated with DMSO or MYCi975 ($2 \mu\text{M}$) with or without knockdown of NME2. Bars represent the cell count quantification of Crystal Violet trapped by migrated cells, normalized to control (DMSO with siScramble). P-value was estimated using 2-way ANOVA, * p-value < 0.05 , ** p-value < 0.01 , *** p-value < 0.001 , **** p-value < 0.0001 . **(G)** Comparison of anti-proliferative effects of Enzalutamide on C42B EnzaRes cells that have NME2 knocked down via CRISPR (sgNME2) versus C42B EnzaRes which have received non-targeting sgRNA control (sgNT). **(H)** Western blot of NME2, MYC and GAPDH protein levels in C42B shNME2 EnzaRes cells treated with Doxycycline for 96 hours at the indicated concentrations. **(I)** Schematic representation and tumor volumes for mice bearing established C42B EnzaRes shNME2 xenografts treated with vehicle ($n = 7$ mice), Enzalutamide (10 mg/kg QD i.p) and/or Doxycycline (2 mg/mL in drinking water) ($n = 6$ mice for Dox, Enza, Enza + Dox arms) for 24 days. Statistical significance performed via two-way ANOVA. * p-value < 0.05 , *** p-value ≤ 0.001 , **** p-value ≤ 0.0001 .

REVIEWERS' COMMENTS

Reviewer #4 (Remarks to the Author):

The authors done great job of addressing all the concerns of this reviewer